

# Dual roles of inorganic aqueous phase on SOA growth from benzene and phenol

Jiwon Choi, Myoseon Jang*, and Spencer Blau

[1]Department of Environmental Engineering Sciences, University of Florida, Gainesville, 32611, USA

*Correspondence to*: Myoseon Jang (mjang@ufl.edu)

**Abstract.** Benzene, emitted from automobile exhaust and biomass burning, is ubiquitous in ambient air. Benzene is a precursor hydrocarbon (HC) that forms secondary organic aerosols (SOA), but its SOA formation mechanism is not well studied. To accurately predict the formation of benzene SOA, it is important to understand the gas mechanisms of phenol, which is one of the major products formed from the atmospheric oxidation of benzene. Our chamber study found that wet-inorganic aerosol

retarded the gas oxidation or phenol or benzene, and thus their SOA formation. To explain this unusual effect, it is hypothesized that a persistent phenoxy radical (PPR) effectively forms via a heterogeneous reaction of phenol and phenol-related products in the presence of wet-inorganic aerosol. These PPR species are capable of catalytically consuming ozone during a $NO_x$ cycle and negatively influencing SOA growth. In this study, explicit gas mechanisms were derived to produce the oxygenated products from the atmospheric oxidation of phenol and benzene. Gas mechanisms include the existing Master Chemical

Mechanism (MCM v3.3.1); the reaction path for peroxy radical adducts originating from the addition of an OH radical to phenols forming low-volatility products (e.g., multi-hydroxy aromatics); and the mechanisms to form heterogeneous production of PPR. The simulated gas products were classified into volatility-reactivity based lumping species and incorporated into the UNIfied Partitioning Aerosol Reaction (UNIPAR) model that predicts SOA formation via multiphase reactions of phenol or benzene. The predictability of the UNIPAR model was examined using chamber data, which were

generated for the photooxidation of phenol or benzene under various experimental conditions ($NO_x$ levels, humidity, and inorganic seed types). The SOA formation from both phenol and benzene still increased in the presence of wet inorganic seed because of the oligomerization of reactive organic species in aqueous phase. However, model simulations show a significant suppression in ozone, the oxidation of phenol or benzene, and SOA growth, compared to those without PPR mechanisms. In addition, the production of PPR is accelerated in the presence of acidic aerosol and this weakens SOA growth. In benzene

oxidation, about 53% of the oxidation pathway is connected to phenol formation in the reported gas mechanism. Thus, the contribution of PPR to gas mechanisms is less than phenol. Overall, SOA growth in phenol or benzene is negatively related to $NO_x$ levels in the high $NO_x$ region (HC ppbC/$NO_x$ ppb <5). However, the simulation indicates that the significance of PPR rises with decreasing $NO_x$ levels. Hence, the influence of $NO_x$ levels on the SOA formation from phenol or benzene is complex under varying temperature and seed types.





## 1 Introduction


Hydrocarbons (HCs) are emitted from both anthropogenic sources (e.g., fuel combustion, vehicle exhaust, and industrial activities) and biogenic sources from vegetation (Carlton et al., 2010). The photochemical oxidation of these HCs can produce ozone through incorporating with the $NO_x$ cycle. In addition, the formation of semi-volatile or non-volatile oxygenated products through a series of chemical reactions of precursor HCs in the atmospheric environment can yield Secondary Organic

Aerosol (SOA). This SOA constitutes a large proportion organic aerosol in the ambient air, ranging from 20% to 90% (Jimenez et al., 2009; Zhang et al., 2007). Hence, the estimation of SOA formation potential is important to accurately evaluate the impact of atmospheric organic aerosol on health and climate formation.

Benzene, the simplest aromatic HC, is emitted mainly from automobile exhaust and also can be found in gases from biomass burning. In addition, benzene, a solvent, is emitted from industrial processes such as chemical synthesis, construction, and

pharmaceutical facilities (Verma and Tombe, 2002; Wang et al., 2014). The major atmospheric oxidation path of benzene is the reaction with OH radicals. Benzene's oxidation rate (i.e., 1.21571E-12 at 298K) is slow, but its SOA yield is high. According to laboratory studies (Ng et al., 2007), benzene's SOA yields can be even greater than high yield aromatic HCs (i.e., toluene) at a given experimental condition. However, the prediction of benzene SOA has not been well studied due to uncertainties in both gas oxidation mechanisms and aerosol phase reactions that form oligomeric species under various

environmental conditions.

In benzene oxidation with an OH radical, about 53% of pathways are linked to the formation of phenol and the remains of oxidation paths are connected to ring opening products in the current Master Chemical Mechanism (MCM 3.3.1) (Bloss et al., 2005). Hence understanding phenol oxidation mechanisms is key to predicting benzene oxidation and SOA formation. Phenol gas oxidation includes the reaction path for peroxy radical adducts originating from the addition of an OH radical to phenols

to form low volatility products (e.g., multi-hydroxy aromatics). Laboratory studies report that gas oxidation of phenol form low-volatility Highly Oxygenated Organic Molecules (HOM) (Nakao et al., 2011; Ji et al., 2017; Yee et al., 2013). Phenols are highly reactive to the addition of OH radicals into an aromatic ring due to the electron receiving characteristics of phenolic OH group. Phenols effectively produce multi-hydroxybenzenes compared to conventional alkyl-substituted monocyclic aromatic HCs (Hansch et al., 2000).

In addition to gas mechanisms, increased SOA formation via the production of HOM, phenol and phenolic products (e.g., catechols and nitrophenols) can yield Persistent Phenoxy Radicals (PPR), which can potentially suppress atmospheric oxidation capability and SOA growth. Unlike aliphatic alkoxy radicals, which can react with an oxygen molecule to form a hydroperoxyl ($HO_2$) radical, PPR have no aliphatic hydrogen at the carbon attached to the O-radical. Thus, the lifetime of phenoxy radicals is relatively long in ambient air and catalytically depletes ozone (Tao and Li, 1999).

In a recent modeling study, Choi and Jang (Choi and Jang, 2022) explicitly predicted phenol gas oxidation including the formation of HOM and other multi-hydroxyphenols (Pillar-Little et al., 2015; Choi and Jang, 2022; Yee et al., 2013; Yu et al., 2016). The resulting oxidation products were categorized into volatility-reactivity based lumping species, and integrated into



the UNIfied Partitioning Aerosol Reaction (UNIPAR) model, which simulated SOA formation via multiphase reactions of HCs. Choi and Jang showed the importance of HOM to predict the formation of phenol SOA through the simulation of chamber

data using the UNIPAR model (Choi and Jang, 2022). In the absence of wet-inorganic seed aerosol, their model successfully simulated SOA formation, but in the presence of wet-inorganic seed, the model failed to predict both gas oxidation (ozone, NOx, and decay of phenol) and SOA formation. In the last two decades, the impact of aerosol acidity on SOA has been studied in numerous laboratories (Jang et al., 2002; Garland et al., 2006; Hallquist et al., 2009). The reactive oxygenated products, formed from the oxidation of biogenic and aromatic HCs, undergo acid-catalyzed reactions (e.g., hydration, oligomerization,

formation of hemiacetal/acetal/trioxane, aldol condensation, and cationic rearrangement) in the presence of acidic inorganic aerosol, and accelerate SOA formation. However, Choi and Jang observed an unexpected impact of aerosol acidity on SOA formation, which suppressed SOA formation more than neutral, wet-ammonium sulfate (AS) seed. Since the wet-inorganic aerosol suppresses gas oxidation (ozone and the decay of phenol), it is unlikely associated with the inaccuracy of oligomerization rates of reactive oxygenated species in aqueous phase. This finding is distinct from the typical SOA formation

from alkyl-substituted aromatics or biogenic HCs, which are positively correlated to aerosol acidity (Jang et al., 2002; Hallquist et al., 2009).

In this study, we hypothesize that the production of PPR from the atmospheric oxidation of phenol and its phenolic products (e.g., catechols and nitrophenols) can be modulated via heterogeneous reactions in wet-inorganic aerosol. The increased PPR via heterogeneous oxidation of phenolic compounds in presence of wet-inorganic aerosol catalytically consumes ozone during

a $NO_x$ cycle, and ultimately influences SOA formation from both phenol and benzene. The gas mechanism, named the Heterogeneous Phenoxy Radical model (H-PPR model), was derived to improve the prediction of both gas oxidation of phenol and benzene and their SOA formation in presence of aqueous phase. The prediction of phenol oxidation was improved by integrating HOM and H-PPR into explicit gas mechanisms. In a similar matter, the benzene gas oxidation was improved by using updated phenol mechanisms. Ultimately the resulting gas mechanisms of phenol or benzene were, then, applied to the

UNIPAR model to predict SOA formation via multiphase reactions of phenol and benzene.[4-9, 52] The suitability of the UNIPAR model was demonstrated by comparing simulation and chamber data obtained from the photooxidation of phenol or benzene under different experimental condition in a large outdoor photochemical reactor.

Importantly, both phenol and benzene are abundant in biomass burning smoke. The SOA model of this study can augment the evaluation of the impact of $NO_x$ levels on SOA formation during wildfires under the rural set (low NOx) and the urban set

(high NOx). The wildfire air plume can transport emissions thousands of km away from the wildfire source (Edwards et al., 2006; Wotawa and Trainer, 2000) influencing background atmosphere air quality (Hudson et al., 2004; Schill et al., 2020). For example, in the Europe Mediterranean area, annually biomass burning emission accounted for 19-21% of organic carbon levels in particulate matter at Barcelona, Spain (Reche et al., 2012). Outdoor chamber data and gas mechanisms of phenols can augment better understanding of the impact of biomass burning smoke on the atmospheric oxidation ability of hydrocarbons

and SOA formation in the city.



## 2. Experiment Section

To generate SOA, the University of Florida Atmospheric PHotochemical Outdoor Reactor (UF-APHOR) was used. The reactor consists of a dual Teflon film chamber, each with a volume of 52 m$^3$ and a surface area of 86 m$^2$, and exposed to ambient sunlight. SOA is produced by the photooxidation of phenol (Acros Organics, 99%) or benzene (Sigma-Aldrich, ≥99%) under varying NO$_x$ levels and different inorganic seed types (non-seed, sulfuric acid (SA), ammonium hydrogen sulfate (AHS) wet-ammonium sulfate (AS)). Detailed description of chamber experiment procedure have been reported in previous studies (Beardsley and Jang, 2016; Choi and Jang, 2022; Han and Jang, 2022, 2023; Im et al., 2014; Yu et al., 2021b; Zhou et al., 2019). In summary, NO (2% in N$_2$, Airgas, USA), HCs, non-reactive CCl$_4$ (Sigma-Aldrich, ≥99.5%), and inorganic seed aerosol were injected into the chamber before sunrise. HCs (phenol or benzene) were vaporized into the chamber using a glass manifold under clean air streams. For seeded SOA experiments, 0.05M SA, AHS, or 0.05M AS aqueous solution was atomized using a nebulizer (LC STAR, PARI, Starnberg, Germany) into the chamber. For dry-AS seeded experiments, the relative humidity (RH) was controlled below efflorescence relative humidity (ERH), and for wet-AS experiments, the RH was maintained above ERH to prevent the crystallization of seed.

A photometric ozone analyzer (Model 106-L, 2B Technologies, MA, USA) and a chemiluminescence NO/NOx analyzer (Model 405, 2B technologies, MA, USA) were used to monitor the concentrations of NO$_x$ and ozone. A Gas Chromatograph with a Flame Ionization Detector (GC-FID) (7820A, Agilent Technologies, CA, USA) was employed to measure the concentrations of phenol or benzene. The Proton Transfer Reaction-Time of Flight (PTR-ToF-MS) (PTR 3C, Kore Technology, Cambridgeshire, UK) was also utilized to monitor the decay of phenol or benzene. To monitor air dilution in the chamber, CCl$_4$ was introduced into the chamber and measured using GC-FID.

The Scanning Mobility Particle Sizer (SMPS) comprising the aerosol classifier (Model 3082, TSI, MN, USA) and condensation particle counter (Model 3750, TSI) was used to measure the particle population and volume concentration. The Aerosol Chemical Speciation Monitor (ACSM) was used to measure the intensity of sulfate, ammonium, nitrate ion peaks in aerosol. An Organic Carbon/Elemental Carbon aerosol analyzer (Sunset Laboratory, OR, USA) was employed to measure the concentration of organic carbon in the aerosol. The Ion Chromatograph (Compact IC 761) was used to measure the concentration of water-soluble inorganic species (Sulfate, nitrate, and ammonium ions) coupled to a Particle into Liquid Sampler (PILS-IC) (ADISO 2081). All gas data were corrected for chamber air dilution. All aerosol data were corrected for gas dilution and the aerosol loss to the chamber wall as performed in previous chamber studies (Beardsley and Jang, 2016; Choi and Jang, 2022; Han and Jang, 2022, 2023; Im et al., 2014; Yu et al., 2021b; Zhou et al., 2019).

A hygrometer (CR1000 measurement and control system, Campbell Scientific, UT, USA) was used to measure temperature and relative humidity (RH) within the chamber. Sunlight intensity was monitored with a Total UV Radiometer (TUVR, Eppley Laboratory, RI, USA). Table 1 provides a summary of the experimental conditions used in the chamber for this study. Fig. S1 displays the profiles of sunlight, temperature, and humidity on October 19, 2022.




# 3. Model Description

## 3.1 UNIPAR SOA model

The UNIPAR model simulates the SOA formation via multiphase reaction of phenol or benzene. Fig. 1 displays the overall structure of the UNIPAR model. The detailed description of the UNIAPR model can be found in Section S2 of Supporting Information. Briefly, the key components of the UNIPAR model are described as follow:

1) In the presence of salted aqueous phase, SOA formation is assumed to be governed by Liquid-Liquid Phase Separation (Choi and Jang, 2022; Han and Jang, 2022, 2023; Im et al., 2014; Yu et al., 2021b; Zhou et al., 2019; Yu et al., 2021a). The

SOA formation in the model streamlines multiphase partitioning, organic-phase (*or*) oligomerization, and aqueous reactions in an inorganic seeded aqueous phase (*in*).

2) As seen in Fig. 1, the oxidized products predicted from near explicit mechanisms of phenol or benzene in gas phase (*g*) are categorized into 50 lumping groups based on their reactivity and volatility. Table S1 displays the physicochemical parameters (i.e., molecular weight ($MW_i$), oxygen-to-carbon ratio ($O:C_i$), and hydrogen bonding ($HB_i$)) of lumping species, which are

used to process multiphase partitioning and aerosol chemistry. The predetermined mathematical equations (Tables S2-S7) dynamically build the stoichiometric coefficient arrays for each precursor (benzene and phenol). Tables S8 and S9 illustrate major products placed in lumping arrays. No difference appears between Table S8 and Table S9 for the major product of each lumping group. The estimated stoichiometric coefficient reflects the influence of $NO_x$ levels and gas aging on gas-product distributions.

3) The concentration of lumping species is distributed into gas ($C_g$), organic ($C_{or}$), and inorganic phases ($C_{in}$) using partitioning coefficients estimated based on Pankow's absorptive partitioning model (Pankow, 1994) with vapor pressure, the estimated activity coefficients of lumping species in both the organic and inorganic phases, and aerosol's average molecular weight in each phase.

4) The resulting $C_{or}$ and $C_{in}$ of each lumping species are applied to process the SOA formation via their multiphase partitioning

($OM_P$) and aerosol phase reactions ($OM_{AR}$) in both *or* and *in*.

5) The kinetic parameters to calculate aerosol phase reaction rate constants in *or* and *in*, such as lumping species' reactivity scales and their basicity constants, are reported Section S2. Both organic-phase oligomerization and aqueous reactions of reactive species in inorganic phase yield non-volatile OM in the model.

6) The SOA mass formed from gas-organic partitioning ($OM_P$) is estimated using the Newtonian method (Schell et al., 2001)

based on a mass balance of organic compounds between the gas and particle phases governed by Raoult's law. $OM_{AR}$ is considered to be a pre-existing absorbing material for gas-particle partitioning (Cao and Jang, 2007; Im et al., 2014).

7) The inorganic composition and aerosol acidity, which are predicted using the inorganic thermodynamic model, are incorporated into the UNIPAR model. The deliquescence RH (DRH), which is predicted using the equation derived from the inorganic thermodynamic model and ERH, which is predicted using a pre-trained neural network model based on the inorganic




composition (Yu et al., 2021a), are used to determine the aerosol state: wet (organic phase + inorganic aqueous phase) or dry (organic phase + solid-dry inorganic phase).

8) In the model, the formation of dialkylsulfate (Liggio et al., 2005; Surratt et al., 2007; Li et al., 2015) is simulated by using the Hinshelwood-type reaction (Im et al., 2014). The decreased acidic sulfate due to the dialkylsulfate formation is applied to inorganic compositions to calculate aerosol acidity and aerosol water content for the next step.

### 3.2 Explicit Gas Mechanisms

The gas oxidation of benzene and phenols was processed with the Master Chemical Mechanism (MCM v3.3.1) coupled with the reaction path for the formation of HOM products (e.g., multi-hydroxy aromatics) and the mechanisms to form PPR via heterogeneous acid-catalyzed reactions of phenolic compounds. The simulation of the atmospheric oxidation of HCs is performed in a box model platform equipped with the Dynamically Simple Model of Atmospheric Chemical complexity

(DSMACC) incorporated with the Kinetic PreProcessor (KPP) (Emmerson and Evans, 2009). The description of the mechanism to form HOM and PPR is shown in the following sections.

#### 3.2.1 HOM Formation

The gas mechanism to form HOM in phenol oxidation has been reported in the recent study by Choi and Jang (2022). Fig. S2 depicts the pathways to form HOM from the oxidation of phenol. The resulting phenol gas mechanisms are integrated with

benzene gas oxidation. The oxidation of phenol or benzene begins with the reaction with an OH radical and forms HOM via multigeneration oxidation (Nakao et al., 2011; Yee et al., 2013; Sun et al., 2010; Garmash et al., 2020; Calvert et al., 2002; Atkinson, 2000; Olmez-Hanci and Arslan-Alaton, 2013). HOM includes multi-hydroxy benzenes, phenolic compounds, and the products derived from peroxy radical adducts of multi-hydroxy benzenes. The reaction rate constants for the addition of the OH radical to different oxygenated products were calculated using the structure-reactivity relationship (Kwok and

Atkinson, 1995).

#### 3.2.2 PPR Formation

The first-generation products from phenol gas oxidation include a phenoxy radical (fraction in oxidation paths: 0.06), catechol (0.657), bicyclic peroxy radical (0.183), and monocyclic peroxy radical (0.1). The resulting phenoxy radical ($C_6H_5O\cdot$) in gas phase can catalytically react with ozone as follows (Tao and Li, 1999),

$C_6H_5OH + OH \rightarrow C_6H_5O\cdot$ (R1)

$C_6H_5O\cdot + O_3 \rightarrow C_6H_5OO\cdot$ (R2)

Phenyl peroxy radical ($C_6H_5OO\cdot$) is able to regenerate $C_6H_5O\cdot$ via reaction with NO, $NO_2$, and $NO_3$ reactions (Carter and Atkinson, 1989; Jagiella and Zabel, 2007) as follow:

$C_6H_5OO\cdot + NO \rightarrow C_6H_5O\cdot + NO_2$ (R3)



$C_6H_5OO\cdot + NO_2 \rightarrow C_6H_5O\cdot + NO_3$                                                                                    (R4)

$C_6H_5OO\cdot + NO_3 \rightarrow C_6H_5O\cdot + NO_2$                                                                                    (R5)

The catalytic decay of ozone influences the production of the OH radical that can be created via the reaction of water vapor with O(1D), a photolysis product of ozone (Finlayson-Pitts and Pitts, 2000). Furthermore, the oxidation of phenol or coexisting HCs can be retarded due to PPR. Evidently, phenol and benzene are known to have a low incremental reactivity to form ozone

(Zhang et al., 2021; Carter, 1994). Fig. 2 illustrates the time series of gas simulations and observations in the UF-APHOR chamber. The gas mechanisms reasonably simulate the low ozone formation and the retarded oxidation of phenol or benzene in the absence of inorganic seed. However, the suppression of gas oxidation capability increased in the presence of wet-AS aerosol and was further amplified with increasing aerosol acidity. This surprising discovery suggests that there should be a reaction path to heterogeneously form PPR.

The formation of phenoxy radicals has been reported in condensed matrixes at low temperatures (Sun et al., 1990), strong liquid acids (Dixon and Murphy, 1976), or gas-phase clusters under specific conditions (Steadman and Syage, 1991). The existence of phenoxy radicals has been detected in strong acidic solutions by using Electron Spin Resonance (ESR) Spectroscopy (Holton and Murphy, 1979). Phenol-like compounds, such as phenol, catechol, and pyrogallol, partition to salted aqueous aerosol and can heterogeneously react with the aqueous-phase OH radical. Various oxidants such as OH, HO$_2$, and

ozone can partition into aqueous phase, and oxidize hydrophilic organic species. Similar to gas phase, the OH radical can be added into an phenolic aromatic ring to form an OH-added intermediate phenol (HO-C$_6$H$_5$·OH) ($phenol\_OH\_int$) in aqueous phase (Mvula et al., 2001). The resulting $phenol\_OH\_int$ can form the phenoxy radical yielding a water molecule (Das, 2005; Mvula et al., 2001). This reaction step is accelerated by an acid catalysis.

In this study, the H-PPR mechanism was integrated into the explicit gas mechanisms accounting for the impact of aqueous

phase on SOA formation from phenol and benzene as illustrated in Fig. 2. In this mechanism, the partitioning of phenols between gas ($g$) and inorganic ($in$) phases is kinetically expressed by using the absorption rate constant ($k_{on\_phenol}$) and the desorption rate constant ($k_{off\_phenol}$) as follows.

$C_6H_5OH(g) \rightarrow C_6H_5OH(in)$          $k_{on\_phenol}$                                                                  (R6)

$C_6H_5OH(in) \rightarrow C_6H_5OH(g)$          $k_{off\_phenol}$                                                                  (R7)

To describe the production of PPR, phenol is used, but various phenolic compounds can also be involved in the formation of H-PPR. $k_{on\_phenol}$ is calculated as follows:

$$k_{on\_phenol} = f_{abs} \frac{\omega\, f_{S\_M}}{4}$$                                                                                          (Eq.1)

$f_{S\_M}$ is the aerosol surface area concentration (m$^2$ m$^{-3}$): i.e., $4\times10^{-3}$, m$^2$ µg$^{-1}$ for the particle size near 100 nm. $f_{abs}$ is the coefficient for the uptake process and set as 2 to gear fast gas-particle partitioning in the model. ω is the mean molecular velocity (m s$^{-1}$)

of each chemical species and is calculated as follows:

$$\omega = \sqrt{\frac{8RT}{\pi MW}}$$                                                                                                              (Eq.2)



MW is the molecular weight (kg mol$^{-1}$) of organic species. R is a gas constant (8.314 J mol$^{-1}$ K$^{-1}$).

In the presence of wet-inorganic seed aerosol, the lumping species produced from the oxidation of HC are split into *g, or* and *in* phases by the gas-aerosol absorptive partitioning model (Pankow, 1994). The *g-in* partitioning coefficient ($K_{in,i}$) (m$^3$ µg$^{-1}$) are expressed as.

$$K_{in,i} = \frac{7.501RT}{10^9 MW_{in} \gamma_{in,i} p°_{l,i}} \quad \text{(Eq.3)}$$

MW$_{in}$ represents the average MW (g mol$^{-1}$) of inorganic phase aerosol. $p°_{l,i}$ is the liquid vapor pressure (in mmHg) of the product *i* and is calculated using the group contribution method (Zhou et al., 2019). The activity coefficient ($\gamma_{or,i}$) of organic species *i* in *or* phase is treated as one (Im et al., 2014). The activity coefficient of *i* in *in* phase ($\gamma_{in,i}$) is predicted using a semi-empirical regression equation. The theoretical estimation of $\gamma_{in,i}$ was conducted by using the thermodynamic Aerosol Inorganic-Organic Mixtures Functional Groups Activity Coefficients (AIOMFAC) (Zuend et al., 2011) for the given set of conditions and aerosol parameters. $\gamma_{in,i}$ is a function of aerosol environment variables (RH ranging from 0 to 1 and fractional sulfate (FS = [SO$_4^{2-}$]/[SO$_4^{2-}$]+[NH$_4^+$])) and the physicochemical parameters of lumping species *i* (MW$_i$, O:C$_i$, and HB$_i$) as follows,

$$\gamma_{in,i} = e^{0.035 \cdot MW_i - 2.704 \cdot \ln(O:C_i) - 1.121 \cdot HB_i - 0.330 \cdot FS - 0.022 \cdot (100 \cdot RH)} \quad \text{(Eq.4)}$$

$k_{off\_phenol}$ is inversely related to the partitioning coefficient ($K_{in,i}$, Fig. 1) of species *i* (phenol) onto the *in* phase with the relation below,

$$k_{off\_phenol} = \frac{k_{on\_phenol}}{K_{in,phenol}} \quad \text{(Eq.5)}$$

The *g-in* partitioning coefficient *($K_{in,ox,}$)* of atmospheric oxidant, *ox* (i.e., H$_2$O$_2$, HONO, HO$_2$, OH, O$_3$, NO, NO$_2$, CH$_3$OOH, and CH$_3$CO$_3$H) is estimated using Henry's constant ($K_{H,i}$) below,

$$\frac{K_{H,ox}}{K_{in,ox}} = 10^6 \frac{M_{in}}{V_{in}RT} \quad \text{(Eq.6)}$$

$M_{in}$ (g/L) is the mass concentration of *in* phase, $V_{in}$ (cm$^{-3}$/L) is volume mixing ratio of *in* phase when R is in 8.205×10$^3$ L atm mol$^{-1}$ K$^{-1}$.

*Phenol (in)* in R6 further reacts with the OH radical *(OH(in))* in *in* phase to form an intermediate adduct of phenol *(phenol_OH_int (in))* at reaction rate constant, $k_1$ as seen below,

C$_6$H$_5$OH(in) + *OH(in)* → *phenol_OH_int (in)*     $k_1$  (7E − 6)     (R8)

Intermediate adduct *phenol_OH_int (in)* can effectively generate a phenoxy radical, C$_6$H$_5$O·*(in)*, via an acid catalyzed reaction below.

$$phenol\_OH\_int\ (in) \xrightarrow{H^+} C_6H_5O·(in) + H_2O \qquad k_{phenoxy} \qquad \text{(R9)}$$

where $k_{phenoxy} = k_1 * e^{[H^+]_{in}}$. $[H^+]_{in}$ is the proton concentration (mol/L) in aqueous phase.

In the similar manner with phenols, the resulting phenoxy radical can also partition between gas and particle phases as follows:

C$_6$H$_5$O·(g) → C$_6$H$_5$O·(in)     $k_{on\_phenoxy}$     (R10)

C$_6$H$_5$O·(in) → C$_6$H$_5$O·(g)     $k_{off\_phenoxy}$     (R11)





The formation of *phenol_OH_int* (*in*) is the rate determining step in the mechanism. In the absence of an acid catalyst, *phenol_OH_int* (*in*) can be further oxidized via the reaction with an oxygen molecule to form catechol and other

multifunctional carbonyls (Xu and Wang, 2013). The produced $C_6H_5O\cdot(g)$ is involved in the reaction with ozone (R2) as seen in Fig. 2 and catalytically reduce atmospheric oxidation capacity.

## 4. Result and Discussion

### 4.1 Simulation of gas oxidation and SOA formation

Fig. 3 (A-L) illustrates gas simulations (phenol, benzene, ozone, NO, and $NO_2$) for chamber data obtained in the UF-APHOR

chamber. In the presence of wet-inorganic seed (i.e., wet-AS, AHS, and SA), both simulations and chamber data show a significant suppression in gas oxidation (i.e., ozone formation and the decay of phenol or benzene) compared to gas oxidation in non-seeded conditions. For example, Fig. 3(B) and (H) display gas simulations excluding the H-PPR mechanism for phenol and benzene oxidation in the presence of SA seed showing a large gap between simulations and observation. Under the same experimental conditions, Fig. 3(C) and (I) illustrate improved gas simulations with H-PPR showing the importance of H-PPR

in accurately predicting the oxidation of phenol or benzene in the presence of wet-inorganic aerosol as discussed in reactions R3-R5. In addition to phenol, catechols and nitrophenols, which are major products from phenol oxidation, can also undergo the PPR formation. The suppressed ozone can lessen the production of an OH radical and further retard the aging of organic products. Overall, explicit gas simulations including HOM and H-PPR agree well with observations.

Fig. 4(A-P) shows chamber-generated SOA mass from the photooxidation of phenol (Fig. 4(A-H)) or benzene (Fig.4(I-P)) in

different inorganic seed conditions (Table 1) and the simulations of SOA formation using the UNIPAR model. Overall, the improved SOA simulation of phenol or benzene was performed with the accurate gas simulation incorporated with HOM and H-PPR. Non-seed phenol SOA is shown in Fig. 4(A) and (B) and non-seed benzene in Fig. 4(I) and (J). SOA masses produced in the presence of inorganic seed (SA, AHS, wet- or dry-AS) are depicted in Fig. 4 (C-H) for phenol and Fig. 4 (K-P) for benzene.

The importance of H-PPR mechanisms on SOA prediction is demonstrated in Fig. 4(C) and (G) for phenol and Fig. 4(K) and (O) for benzene by comparing the simulations with H-PPR and without H-PPR. The suppression of SOA formation was greater with highly acidic aerosol. Additionally, the formation rate of PPR can be influenced by the chemical composition of the aerosol medium. Mitroka et al. (Mitroka et al., 2010) reported that reactivity of the OH radical is considerably higher in polar, protic solvent than that in dipolar, aprotic solvent. Protic solvent is a hydrogen bond donor that stabilizes the transition

state of the OH radical addition reaction. Thus, the reaction of phenols with the OH radical is more favorable in *in* phase than *or* phase. The radical scavenging ability of phenols by forming phenoxy radicals is in the order of pyrogallol > 1,2,4-benzenetriol >catechol > hydroquinone > resorcinol ≈ phloroglucinol (Thavasi et al., 2009). As shown in reaction R8, phenol in salted aqueous media reacts with $OH$ (*in*) in a similar way with the OH addition to the aromatic ring in the gas phase to form intermediate product *phenol_OH_int* (*in*) (Fig. 2). Fig. S3 is the proposed mechanism to form phenoxy radical via the acid-




catalyzed reaction. In addition, some organic products such as quinones can promote increased oxidants in aqueous acidic media. Quinones are well recognized for their ability to promote superoxide formation (Guin et al., 2011). Lowering pH increases the redox potential (Walczak et al., 1997) of quinone-hydroquinone. However, the reduction potential of oxygen can be lower in acidic condition and is advantageous for $O_2^{\cdot-}/HO_2\cdot$ formation (Wei et al., 2022) (Section S4).

The importance of HOM on phenol SOA has been demonstrated in the previous study by Choi and Jang (Choi and Jang, 2022).
For example, a large fraction of $OM_P$ in Fig. 4(A) is contributed by HOM. The contribution of HOM to SOA mass increases with decreasing NO levels. The systematic evaluation of the UNIPAR model integrated with the explicit gas mechanisms will be performed via the model sensitivity to various environmental variables (i.e., $NO_x$ levels, seed, temperature, and humidity) in section 4.2.

**4.2 Sensitivity of SOA Formation to Environmental Variables**

**4.2.1 Evaluation of the impact of H-PPR on SOA Formation: aerosol acidity**

In order to assess the impact of H-PPR on phenolic SOA and benzene SOA, the UNIPAR simulation was performed as a function of aerosol acidity (FS value) as seen in Fig. 5. This sensitivity was performed under the given conditions: temperature = 298K, $NO_x$ level = 7.5 ppbC/ppb, and RH = 0.6 at the sunlight conditions on October 19, 2022 (Fig. S1). Under this RH, all seeds are wet. The concentration of the initial HCs for each simulation was set to 30 ppb at four different HC compositions in
Fig. 5: (A) phenol; (B)benzene; (C) phenol:benzene = 3:1 ppb/ppb, and (D) phenol:benzene = 1:1 ppb/ppb. The concentrations of consumed phenol or consumed benzene varied according to the different rate constant of each HC with the OH radical and the H-PPR degree.

In Fig. 5, a considerable difference was identified between the SOA mass simulated with H-PPR (solid line) and that without H-PPR (dashed line). SOA growth is suppressed with H-PPR. This gap gradually increases with increasing aerosol acidity
(FS). The reduction in SOA mass with acidic seed mainly is caused by the retardation of gas oxidation, which is linked to the catalytic consumption of ozone, which is consequently connected tothe slow oxidation of phenol and slow aging of oxygenated products (i.e., Figs 3B and 3C) or benzene (i.e., Figs 3H and 3I). The simulation also depicts the impact of the amount of inorganic seed on SOA formation of phenol or benzene. Surprisingly, phenol SOA mass is lower with the 30 μg/m³ of inorganic seed than that with 5 μg/m³ of inorganic seed, in contrast to the typical tendency in the SOA formation from non-
phenolic HCs..

Compared to phenol (Fig. 5(A)), the impact of H-PPR on benzene SOA formation is small as seen in Fig. 5(B). Benzene is more affected by acid-catalyzed oligomerization than phenol. Benzene oxidation yields various products other than phenol. Some reactive organic products are involved in oligomerization in aqueous phase. As seen Fig. 5(C) and (D), when benzene is mixed with phenol in the presence of the high concentration of wet seed, SOA growth shows more suppression suggesting
the impact of PPR from phenol. The SOA mass difference due to the H-PPR mechanism is large in the mixture of phenol and benzene. The smaller amount of phenol consumption in the gas mixture as seen in Fig. 5(C) and (D) yields less SOA mass,





which impacts partitioning of products and their oligomerization in aerosol. Thus, the impact of H-PPR on SOA mass in Fig. 5(C) and (D) is relatively more significant than that in Fig. 5(A) due to the difference in SOA yields. These simulation results suggest that phenolic compounds in biomass burning smoke gases might impact the atmospheric oxidation ability of urban air.

**4.2.2 Sensitivity of SOA formation to NOₓ level, Temperature, and RH**

Fig. 6 (6A-6D) displays the sensitivity of the SOA mass to $NO_x$ levels at three different temperatures (278K, 288K and 298K) and two different seed conditions (no seed and AHS seed at RH = 0.6). Fig. 7 illustrates the sensitivity of phenol or benzene SOA mass to RH (0.3 and 0.6) at two different seed conditions (AHS and AS seed). The UNIPAR simulation was evaluated at the given sunlight conditions on October 19, 2022 (between 6:30 AM to5:30 PM EST, Fig. S1). The initial concentration of

HCs in each simulation in Figs. 6 and 7 was set to 30 ppb.

A typical negative correlation appears between SOA mass and temperature (Fig. 6). Regardless of seed conditions, a typical $NO_x$ impact on SOA formation appears at high $NO_x$ regions (HC ppbC/$NO_x$ ≤ 5) showing a negative correlation between $NO_x$ levels and SOA formation. At the higher $NO_x$ level, more organonitrate forms via the reaction of peroxy radicals with NO. Organonitrates attribute to SOA formation mainly via the partitioning process and their volatility is relatively high (Choi and

Jang, 2022). Phenol SOA is more sensitive to the $NO_x$ level than benzene SOA. As the $NO_x$ level decreases in low $NO_x$ regions (HC ppbC/$NO_x$ > 5), SOA production gradually increases because of the increased contribution of low-volatile multi-hydroxyl products (HOM products that form via multiple additions of OH radicals into phenol) in the absence of wet seed (Fig. 6(A) and 6(C). Thus, the increased HOM products at the low $NO_x$ level decrease the sensitivity of SOA formation to temperature. In the presence of acidic seed, the production of phenoxy radicals increases with decreasing $NO_x$ levels due to

increased oxidants in aerosol phase, and negatively influences SOA formation.

Phenol rapidly reacts with OH radicals (2.82E-11 cm³/molecule s⁻¹ at 298K) in the gas phase and quickly produces low volatile products compared to benzene (1.2E-12 cm³/molecule s⁻¹). Thus, phenol produces a higher SOA mass than benzene at given simulation conditions (at the same initial HC concentration, $NO_x$ levels, and sunlight). The reaction rate of benzene with the OH radical is even slower than that of $NO_2$ with the OH radical. Nearly 47% of benzene oxidation would form gas-oxidation

products other than phenol and some of them can be involved in aerosol phase oligomerization. Benzene SOA yields can be significant although benzene decay is slow (Fig. 3 and Fig. 4).

AS seed in Fig. 7(A) can be effloresced at RH 0.35-0.4. Thus, AS-seed is dry at RH= 0.3 but wet at RH = 0.6. Fig. 7(A) display a higher SOA formation at 0.6 RH than 0.3 RH because of aqueous phase oligomerization(Choi and Jang, 2022; Han and Jang, 2022, 2023; Im et al., 2014; Yu et al., 2021b; Zhou et al., 2019; Yu et al., 2021a). Overall, both SOA are slightly

sensitive to RH conditions. In the presence of AHS seed that is wet in both RHs (0.3 and 0.6), both phenol and benzene exihibit an unusual tendency showing higher SOA mass with lower RH. SOA growth from both benzene and phenol can increase due to acid-catalyzed oligomerization but it can be partly suppressed via the H-PPR mechanism. Fig. 7(B) and 7(C) display $OM_{AR}$ (oligomeric SOA mass) and $OM_P$ (partitioning mass) contributions to $OM_T$. Both benzene SOA and phenol SOA are dominated by $OM_{AR}$ at given simulation conditions.





### 4.2.3. Uncertainty of SOA Formation to Model Parameters

Fig. 8 illustrates the uncertainties of important model parameters in predicting SOA mass using the UNIPAR model. The partitioning process ($OM_P$) is most impacted by the lumping species' vapor pressure (VP). $OM_{AR}$ is influenced by the reaction rates of oligomerization of the reactive lumping species ($k_{oligomerization}$) and H-PPR ($k_{phenoxy}$). The uncertainties of the estimated VP are set to the reported value associated with the group contribution method (Zhao et al., 1999; Yu et al., 2021b). The corresponding change in the SOA mass owing to VP uncertainties ranges from -20% to 24% in benzene SOA simulation with wet-AS seed at given simulation conditions (Fig. 8(A)). Benzene SOA (Fig. 8(A)) is more sensitive to the VP uncertainty than phenol SOA (Fig. 8(B)). As seen in Fig. 8C and 8D, SOA mass is more significantly impacted by the uncertainty associated with the oligomerization rate constant than that of the H-PPR rate constant at a given uncertainty range. The variation in SOA formation with the change of $k_{phenoxy}$ is trivial. For H-PPR, the amount of available oxidants (i.e., OH radicals, HONO, and $H_2O_2$) is more critical than $k_{phenoxy}$ in our model.

### Conclusion and atmospheric implications

Hitherto, the typical role of inorganic acids was known to be an acid catalyst that can accelerate the oligomerization of reactive organic species (i.e., aldehydes and epoxide) in aerosol phase (Jang et al., 2002; Garland et al., 2006; Hallquist et al., 2009). SOA growth accelerated by an acid catalyst has been reported in various chamber studies including both aromatic and biogenic HCs. Surprisingly, the acidic aerosol along with phenol and benzene suppressed the oxidation of air including ozone formation and the decay of HCs (Fig. 3) due to the heterogeneous formation of PPR.

The gas mechanisms including formation of HOM and H-PPR simulated well gas oxidation (i.e., ozone, $NO_x$ and the decay of benzene or phenol) and improved the prediction of SOA formation using UNIPAR (Fig. 4). As seen in Fig. 5, phenol SOA still increases via heterogeneous oligomerization showing a difference in SOA mass between dry- and wet-AS seed. As discussed in Section 3.2.2, the heterogeneously produced PPR production occurs via the reaction of phenolic compounds with the aerosol-phase OH radical that is available during daytime due to the photolysis of $H_2O_2$ and HONO. Thus, the heterogeneously produced PPR is effective with wet-inorganic aerosol during the daytime.

Fundamentally, biomass burning under open flame is performed at low temperature and produces very low NO (Simoneit, 2002; Mebust and Cohen, 2013; Xu et al., 2021). The chemistry slows to a standstill without $NO_x$ and thus halts ozone formation although gaseous HCs are abundant. When these fire plumes mix into urban atmospheres abundant in $NO_x$, ozone formation becomes active, impacting the air quality of the city. The results from this study suggest that PPR produced during the atmospheric process of phenolic compounds in wildfire plumes can retard the atmospheric oxidation in urban environments. Importantly, there are numerous phenolic compounds in biomass burning smoke. Phenolic compounds including phenol, cresol, catechol, methoyphenols, dimethylphenols (Akherati et al., 2020; Bruns et al., 2016) can consist of more than 80% of the precursor HCs in wildfire smoke. These multifunctional phenolic compounds can yield PPR as active scavengers for ozone (Section 3.2.2). In particular, hygroscopic inorganic aerosols comprising of nitrate, sulfate and



ammonium ions are available in the city environment rich in $NO_x$, $SO_2$ and $NH_3$. When wildfire plumes mix in city air, concentrations of their phenolic compounds are diluted. Unlike the laboratory chamber study, atmospheric oxidation occurs with much lower concentration of HCs including phenolic compounds. As discussed in Fig. 5, the impact of H-PPR on SOA

formation is relatively much more significant with the lower concentration of HCs. To date, the impact of phenolic compounds on retardation of atmospheric aging of HCs in the city air has not been sufficiently studied. It is important to comprehend the formation mechanisms of PPR-like chemical species and their role on atmospheric oxidation capability to accurately predict the elevation of ozone and SOA and their peak time.

The impact of $NO_x$ on SOA formation appeared to be negative as shown in Fig. 6 under high $NO_x$ levels. A significant fraction

of phenolic SOA is through HOM products and oligomeric matter. The contribution of HOM and oligomeric matter on SOA formation is generally higher with lower $NO_x$ levels. Thus, phenol SOA and benzene SOA are relatively insensitive to temperature (Fig. 6) due to the high fraction of SOA mass being non-volatile. This result suggests that SOA from biomass burning is not substantially affected by temperature under low $NO_x$ regimes. The role of phenolic compounds on atmospheric oxidation capacity needs to be studied under different $NO_x$ levels to identify atmospheric chemistry in urban and rural sets.

Several unresolved issues need to address to accurately predict SOA formation from biomass burning smoke, including gas mechanisms of unidentified HCs in wildfire smoke, the role of the aqueous phase on the formation of PPR, and unidentified SOA formation mechanisms. For example, there are still missing mechanisms, including cross-reactions of $RO_2$ radicals and the distribution of oxidation paths. This study focused on the oxidation of phenol and benzene through their reaction with OH radicals. Ozone, which mainly forms during daytime, often persists through the night and reacts with $NO_2$ to form nitrate

radicals at night. Phenol can react with this nitrate radical during the night although its reaction is slower than that with OH radical. The reaction of phenols with nitrate radical can also produce PPR-like species and can catalytically scavenge ozone. Nighttime humidity, which increases as temperature decreases, can surpass delinquent RH (DRH) of hygroscopic inorganic constituents and form wet inorganic aerosol. The resulting aerosol is wet until humidity drops below ERH. To better predict SOA formation, the atmospheric processes of biomass burning smoke needs to be studied under different climate conditions,

such as temperature, humidity, and sunlight, and emissions of air pollutants.

Phenol is the most abundant first-generation product from the oxidation of benzene (Smith et al., 2014; Johnson et al., 2004). Additionally, the formation of cresol is involved in 18% of toluene-OH reactions. Xu et al. explained that toluene SOA formation was underestimated by about 20% proposing the uncertainty in the formation of phenolic compounds (Xu et al., 2015). Dimethylphenols can also be produced from the oxidation of xylenes. Comprehension of the atmospheric process of

phenolic compounds can improve the prediction of SOA formation from aromatic HCs. A large fraction of phenolic compounds is directly emitted but its impact on SOA mass is missing in the current air quality model (Pye et al., 2023).



## Acknowledgments

This research was supported by the National Institute of Environmental Research (NIER2020-01-01-010), the National Science Foundation (AGS1923651), and the Fine Particle Research Initiative in East Asia Considering National Differences (FRIEND)
Project through the National Research Foundation of Korea (NRF) funded by the Ministry of Science and ICT (2020M3G1A1114556).

Data availability. Chamber simulation data is available upon request.

Author contributions. JC and MJ conducted chamber experiments and simulated the UNIPAR SOA model. JC, MJ, and SB processed chamber data.

Competing interest. The authors declare that they have no conflict of interest.

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





**Table 1.** Chamber experimental conditions for the SOA formation from the photooxidation of phenol and benzene under varying conditions.

| HC | No. | Date | [a]Initial HC ppb | Initial NO$_x$ ppb | Initial HONO ppb | [b]HC/NOx ppbC/ppb | [c]Seed | Seed mass µg/m3 | RH % | Temp K | [d]ΔHC µg/m3 | [e]SOA Mass µg/m3 | Yields | comments |
|---|---|---|---|---|---|---|---|---|---|---|---|---|---|---|
| Phenol | 1 | 090721 | 79 | 85 | N/A | 5.6 | No seed | N.A. | 19-45 | 298-320 | 200 | 71 | 0.35 | Fig. 4 |
| | 2 | 020723 | 227 | 104 | N/A | 13.1 | No seed | N.A. | 26-94 | 281-311 | 821 | 164 | 0.20 | Fig. 3 Fig. 4 |
| | 3 | 090721 | 130 | 90 | N/A | 8.0 | SA | 332 | 15-56 | 296-320 | 185 | 48 | 0.25 | Fig. 4 |
| | 4 | 020723 | 264 | 74 | N/A | 21.4 | SA | 129 | 25-89 | 281-308 | 546 | 164 | 0.30 | Fig. 3 Fig. 4 |
| | 5 | 040623 | 92 | 59 | N/A | 9.4 | AHS | 2201 | 23-89 | 292-320 | 330 | 204 | 0.62 | Fig. 3 Fig. 4 |
| | 6 | 040623 | 89 | 303 | N/A | 1.8 | AHS | 277 | 29-94 | 293-320 | 319 | 183 | 0.57 | Fig. 4 |
| | 7 | 120222 | 148 | 54 | N/A | 16.4 | d-AS | 62 | 21-44 | 282-309 | 300 | 160 | 0.53 | Fig. 3 Fig. 4 |
| | 8 | 120222 | 162 | 62 | N/A | 15.6 | w-AS | 141 | 59-98 | 282-309 | 319 | 179 | 0.56 | Fig. 3 Fig. 4 |
| Benzene | 9 | 061722 | 292 | 32 | 75 | 16.4 | No seed | N.A. | 36-98 | 296-319 | 208 | 39.6 | 0.19 | Fig. 3 Fig. 4 |
| | 10 | 061722 | 168 | 191 | 101 | 3.44 | No seed | N.A. | 24-84 | 296-322 | 163 | 13.5 | 0.08 | Fig. 3 |
| | 11 | 080722 | 325 | 110 | 47 | 12.4 | SA | 372 | 21-56 | 297-319 | 112 | 29.4 | 0.26 | Fig. 3 Fig. 4 |
| | 12 | 080722 | 310 | 303 | 74 | 4.9 | SA | 377 | 27-57 | 298-316 | 153 | 36.3 | 0.24 | Fig. 4 |
| | 13 | 092022 | 266 | 65 | 77 | 11.2 | AHS | 95 | 27-93 | 295-320 | 134 | 32.9 | 0.25 | Fig. 3 Fig. 4 |
| | 14 | 092022 | 233 | 341 | 71 | 3.39 | AHS | 153 | 38-99 | 295-318 | 112 | 20.4 | 0.18 | Fig. 4 |
| | 15 | 121822 | 270 | 26 | 74 | 16.2 | d-AS | 10 | 27-54 | 281-301 | 125 | 42.6 | 0.34 | Fig. 3 Fig. 4 |
| | 16 | 121822 | 238 | 21 | 61 | 17.4 | w-AS | 71 | 29-88 | 277-300 | 185 | 44.5 | 0.24 | Fig. 3 Fig. 4 |

620    N.A.: not applicable

a.  High NO$_x$: HC ppbC/NO$_x$ ppb < 5 ; low NO$_x$: HC/NO$_x$ > 5 ppbC/ppb
b.  The seed condition refers to the injected electrolytic seed: sulfuric acid (SA), ammonium hydrogen sulfate (AHS), wet ammonium sulfate (wet-AS), and dry AS (dry-AS).
c.  ΔHC is difference between initial HC concentration and the HC concentration at 4:00 PM EST.
625    d.  The SOA mass is determined with OC data at 4:00 PM EST. The reported SOA mass was corrected for the particle loss to the chamber wall based on the 1$^{st}$ order deposition rate at 64 particle size beans.





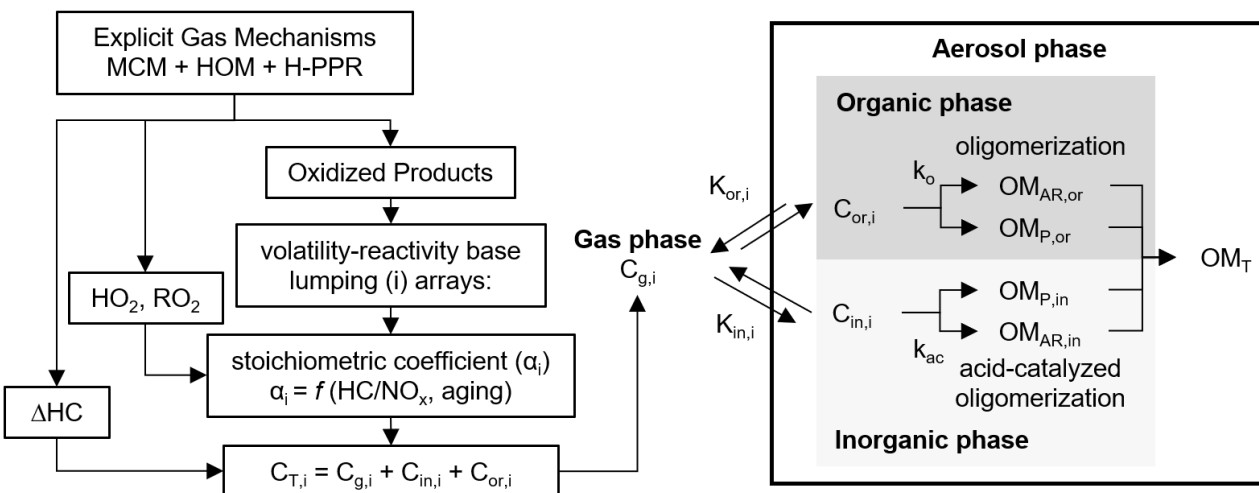

Figure 1: Scheme of the UNIPAR model to predict the SOA formation from the multiphase oxidation of phenol or benzene. The oxidized products predicted from the modified explicit gas mechanism (MCM v.3.3.1) integrated with the H-PPR model are classified into 50 lumping species (i) based on volatility and reactivity. The consumption of hydrocarbons (ΔHCs), the concentration of hydroperoxy radical (HO2), alkylperoxy radical (RO2), and the organic products are also simulated by using MCM and applied to the UNIPAR model. The lumping array associated with stoichiometric coefficients is dynamically constructed as a function of the HC ppbC/NOx ratio and the aging scale, which is estimated with the concentrations of HO2 and RO2 radicals. "C" denotes the concentration of an organic compound and K denotes partitioning coefficient of an organic compound. Subscripts "g", "or", and "in" represent gas, organic, and inorganic phases, respectively. OM refers the organic matter in aerosol. Subscripts "AR", "P", and "T" refer aerosol-phased reaction, partitioning, and total.





640 Figure 2 : The overview of the kinetic mechanisms, which streamlines the heterogeneous phenoxy radical formation (H-PPR) of gaseous phenols in the presence of acidic aerosol. $k_{on}$ and $k_{off}$ denote the uptake rate constant and the desorption rate constant, respectively for phenols in the presence of acidic aerosol. k1 is the rate constant to heterogeneously form intermediates (hydroxy-phenols) and $k_{phenoxy}$ is the rate constant to from an intermediate to a phenoxy radical.





Figure 3 : The time profiles of observations and the prediction for concentrations of NO, NO2, and O3 and hydrocarbons (Table 1). "HC" and "HC_exp" demote the gas simulation of hydrocarbons used in experiment and measurement of hydrocarbon used in experiment, respectively. The error associated with NO, NO2, and O3 are 2% and not visible in this Figure.







Figure 4 : The simulated SOA mass in the different seed condition and measured SOA mass in the different seed condition under different NOx. Solid lines indicate the simulated SOA mass and bullet points (•) indicate the experimentally measured mass. The SOA mass simulated with the H-PPR model (red solid line) was compared with that simulated without the H-PPR model (blue solid line). The UNIPAR-predicted $OM_{AR}$ (heterogeneous reaction in aerosol phase) and $OM_p$ (partitioning) are also included. The error associated with SOA data was 9% according to the uncertainty in OC/EC data.



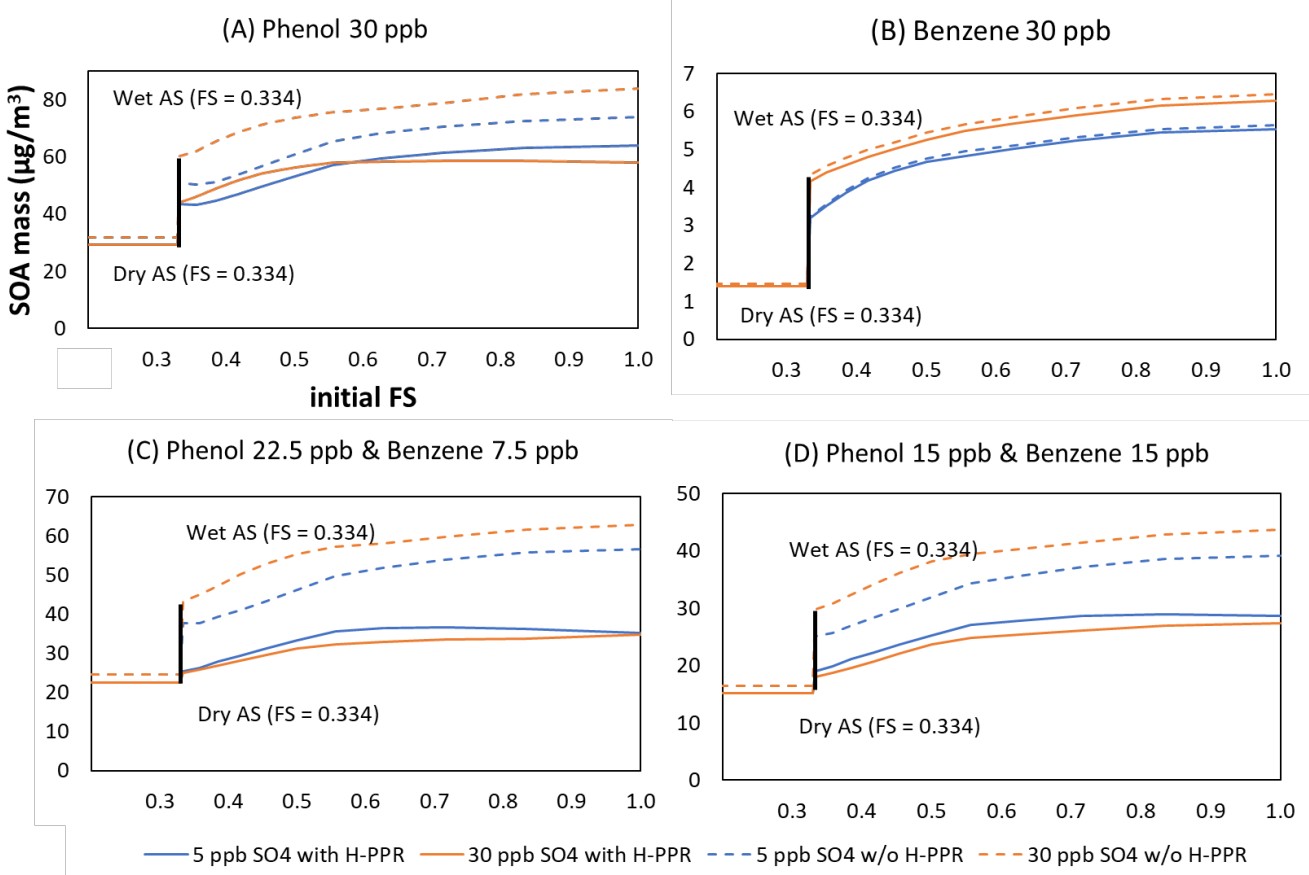

**Figure 5.** Sensitivity of SOA production to aerosol acidity (FS value) for different precursor mixes using the UNIPAR model: phenol (A), benzene (B), phenol:benzene = 3:1 ppb/ppb (C), and phenol:benzene = 1:1 ppb/ppb (D). The initial precursor hydrocarbon concentrations were 30 ppb. The initial concentrations of HONO, formaldehyde, acetaldehyde were 5 ppb, 6 ppb, and 2ppb, respectively. The simulation was performed under ambient sunlight on October 19, 2022. RH was set to 0.6, temperature was to 298K, and HC ppbC/$NO_x$ ppb was 6. 5 ppb sulfate = 20 $\mu g/m^3$ and 30 ppb sulfate = 120 $\mu g/m^3$.



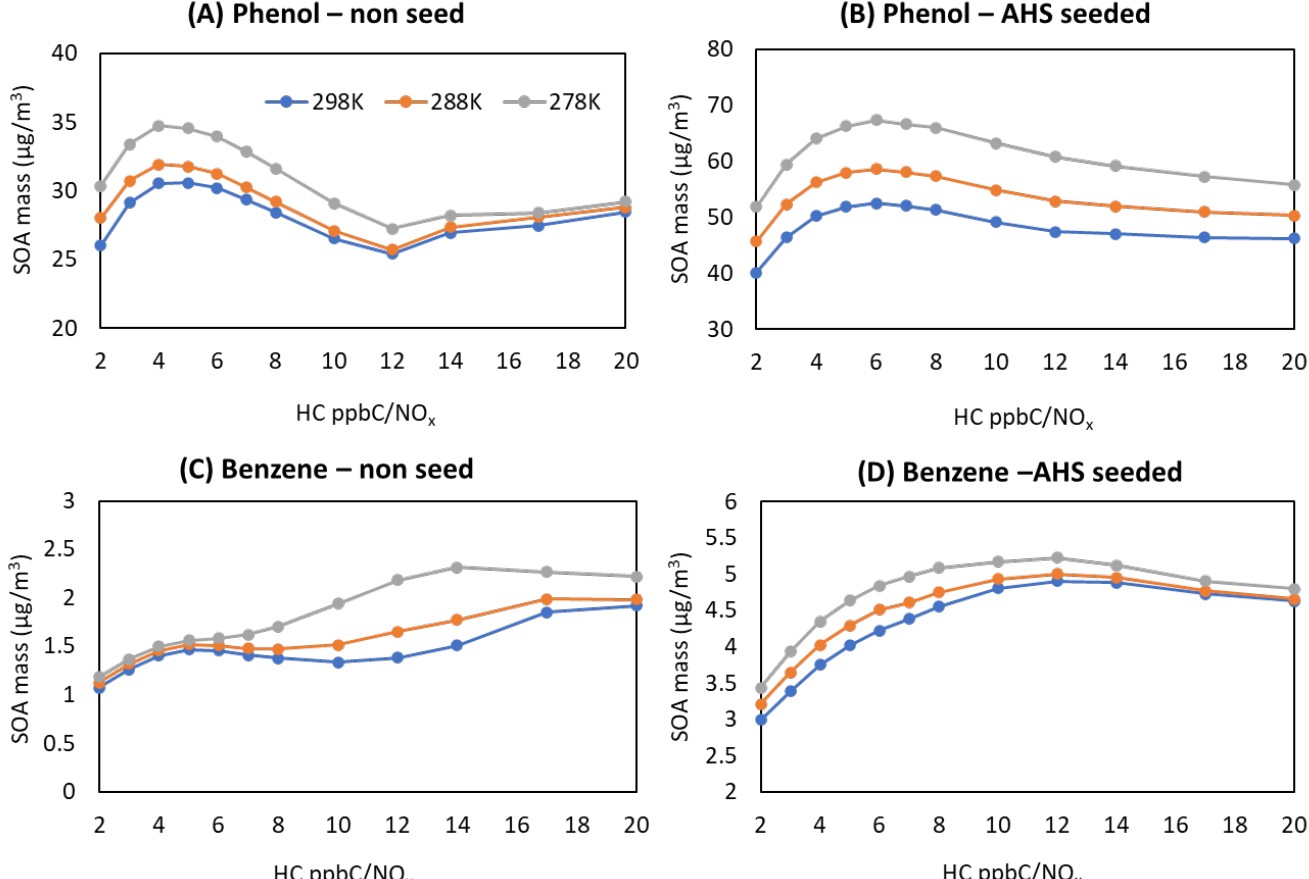

**Figure 6**. Sensitivity of SOA mass to major model variables. SOA simulations were performed at a given sunlight profile on October 19, 2022 (from 6:30 to 17:30, local time). All simulations were performed with 30 ppb of initial HC concentration. The predicted phenol SOA mass to $NO_x$ levels three different temperature in the presence of ASH seed (A); the predicted phenol SOA mass to $NO_x$ levels three different temperature in absence of seed (B); the predicted benzene SOA mass to $NO_x$ levels three different temperature in the presence of ASH seed (C); and the predicted benzene SOA mass to $NO_x$ levels three different temperature in absence of seed (D). Temperature levels were set to 298 K, 288 K, and 278 K. The NOx levels (HC ppbC/NOx ppb) range from 2 to 20.



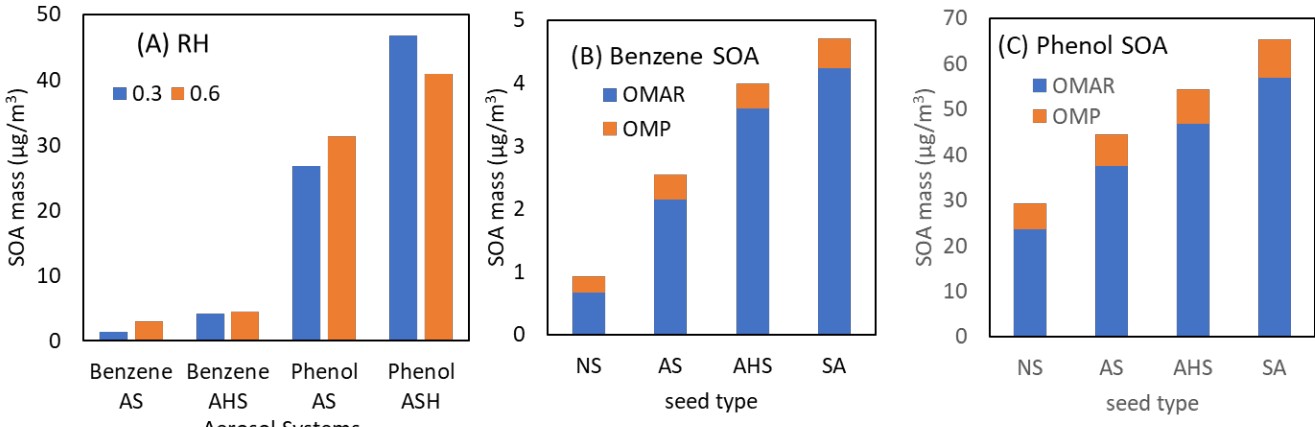

Figure 7. (A) The predicted SOA mass from the photooxidation of benzene or phenol at two different RH (0.3 and 0.6) in the presence of AS or AHS seed (HCpppbC/NO$_x$ = 6.64). OM$_{AR}$ (oligomeric mass) and OM$_P$ (partitioning mass) in benzene SOA (B) and phenol SOA (C).  For (B) and (C), SOA production is simulated with initial HC concentration = 30 ppb (HC ppbC/NO$_x$ = 6.64), RH= 0.6, temperature = 298K, and sulfate = 20 µg/m$^3$ under the sunlight condition on 10/19/2022.





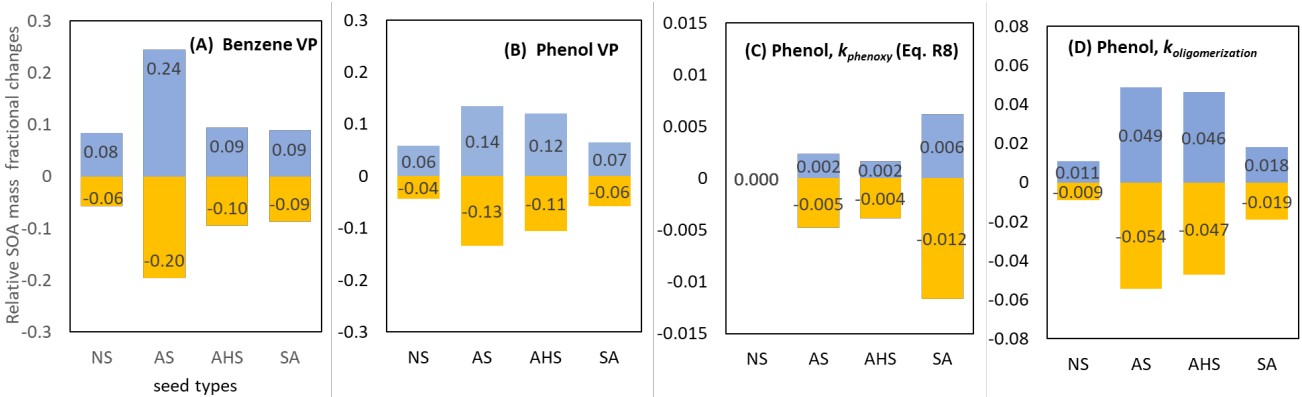

Figure 8. The relative change in SOA mass from the photooxidation of benzene or phenol. The variation in VP of benzene (A) and phenol (B), $k_{oligomerization}$ of phenol SOA (C) and $k_{phenoxy}$ of phenol SOA (D) is set to the factor of 0.5 and 2. SOA production is simulated with initial HC concentration = 30 ppb (HC ppbC/NO$_x$ = 6), RH= 0.6, temperature = 298K, and sulfate

685 = 20 µg/m$^3$ under the sunlight condition on 10/19/2022.