# Peer review of "Dual roles of inorganic aqueous phase on SOA growth from benzene and phenol"

_EGUsphere, 2023_

## Referee Comment (RC1)

The manuscript "Dual roles of inorganic aqueous phase on SOA growth from benzene and phenol" provides new insight into the SOA formation processes from the oxidation of gaseous benzene and phenol under various HC:NOx ratios. To date, experimental studies show a negative related NOx dependence of SOA formation yield from the oxidation of aromatic hydrocarbons. The work presented herein combines experimental chamber investigations with a complex modeling system to deeply explore the heterogenous chemistry within SOA particles with respect to various relevant environmental parameters (i.e., acidity of SOA particles, SOA thermodynamical equilibrium, partitioning coefficients, temperature and RH). Authors employed a variety of modeling tools and used available atmospheric databases (MCM, EPI Suite) to design a tool for predicting the SOA mass under different atmospheric conditions by mean of heterogenous reactions in a two media particle system (inorganic/organic liquid phases) and by gas-particle partitioning processes. Acid-catalyzed formation of a persistent phenoxy radical (PPR) in wet inorganic aerosols and its desorption into the gas phase is hypothesized to be responsible for ozone consumption, thus lowering the atmospheric oxidation capacity near human settlements. Significant improvements were made to the in-use UNIPAR model by integrating HOM and H-PPR sequences to accommodate a new gas mechanism driven by the oxidation of benzene and phenol.

Both the experimental and the modeling part are well presented through the manuscript. I recommend this manuscript for publication in ACP after the following concerns are addressed.

Initial manuscript evaluation: Major revisions

You may be more explicit in the abstract about the "Dual roles of inorganic aqueous phase". For instance, "Data presented herein highlights the impact of aqueous phase on SOA generated through benzene and phenol oxidation. The roles of the aqueous phase consist in: (1)…. and (2)… .

A discussion regarding minimal incremental reactivity index (MIR) (Carter, 1994/ https://doi.org/10.1080/1073161X.1994.10467290) and photochemical ozone creation potentials (Jenkin et al., 2017/ https://doi.org/10.1016/j.atmosenv.2017.05.024) of monocycle aromatics would add considerable impact to your current findings and highlight the atmospheric implications.

To what extent would the competing reaction of PPR with the dissolved $NO_2$ in the inorganic phase affect the UNIPAR/H-PPR model (Kleffmann et al., 1998/ https://doi.org/10.1016/S1352-2310(98)00065-X)? Same question for the catechol gas-phase reactions with ozone (Obeid et al., 2024/ https://doi.org/10.1016/j.envpol.2023.122743; Coeur-Tourneur et al., 2009/ https://doi.org/10.1016/j.atmosenv.2008.12.054; Thomas et al., 2003/ https://doi.org/10.1002/kin.10121)

How is $k_{off\_phenoxy}$ calculated? Is it assumed to be equal to $k_{off\_phenol}$? If so, explain why and how an order of magnitude in between the considered value impact the model? Does the model incorporate Leighton equilibrium in predicting the gas-phase $O_3$, $NO_2$ and NO concentrations?

Kwok and Atkinson SAR on monocyclic aromatics follows the regression log ($k$/cm$^3$ molecule$^{-1}$ s$^{-1}$) = -11.6 − 1.39 $\Sigma\sigma^+$, where $\sigma^+$ are the Hammett constants for electrophilic substitution by Brown and Okamoto (1958/ https://doi.org/10.1021/ja01551a055). If you are using EPI Suite software to estimate the gas kinetic rate coefficients for multi-hydroxy benzenes with vicinal OH groups the software may underestimate the values (Roman et al., 2022/ https://doi.org/10.5194/acp-22-2203-2022).

Also, you could calculate and provide in the discussions sections a relative drop in $NO_2$, $O_3$ and SOA mass concentration when applying the UNIPAR with and without H-PPR.

Using the current dataset for the UNIPAR/H-PPR, could you estimate the SOA mass distribution from the oxidation of 2-methylphenol and catechol under similar conditions?

Technical corrections, minor questions and suggestions:

Affiliation is not indicated for the authors.

Abstract

L10: gas oxidation or phenol or benzene… > gas oxidation of phenol or benzene…

L25: oxidation, about 53% of the… > oxidation, up to 53% of the…

Across the manuscript you have no consistency expressing the units (i.e., L227: g mol$^{-1}$, L241: g/L). Choose one way to express the units.

Introduction

L41: oxidation rate (i.e., 1.21571E-12 at 298K) > oxidation rate (i.e., $1.22 \times 10^{-12}$ cm$^3$ molecule$^{-1}$ s$^{-1}$ at 298K) [REFERENCE NEEDED]. Be consistent with the units and the order of magnitude across the manuscript and the supplement material.

L41: but its SOA yield is high > [provide a range for observed SOA formation yield and the corresponding cited paper/ papers].

L59: The lifetime is long also due to a p-π conjugated system also help for stabilizing the phenoxy radicals.

L85: delete "[4-9, 52]".

L86: of phenol or benzene > of phenol and benzene

Experiment section

L 109: Specify the instrument and the operating conditions used to monitor the HCs concentration presented in Fig 3. What were the sensitivities and the corresponding relative uncertainties for NO/NOx (Villena et al., 2012/ https://doi.org/10.5194/amt-5-149-2012) and O$_3$ (Spicer et al., 2012/https://doi.org/10.3155/1047-3289.60.11.1353) photometers? In what extent these uncertainties would affect the experimental findings?

L 116: Regarding the SOA seeds, were particle diameters the same for all experiments? Do you account for differences in SOA surface concentration in the UNIPAR model?

L 117: sulfate, ammonium, nitrate ion peaks in aerosol. > sulfate, ammonium and nitrate ion signals in aerosol phase.

L 120: species (Sulfate, nitrate… > species (sulfate, nitrate…

UNIPAR SOA model

L153: You stated that "Both organic-phase oligomerization and aqueous reactions of reactive species in inorganic phase yield non-volatile OM in the model". Except for PPR, right?

HOM Formation

L178: The reaction rate constants > The gas phase reaction rate coefficients

PPR Formation

L183: A citation is needed for the branching ratios.

L222: R is a gas constant (8.314 J mol$^{-1}$ K$^{-1}$). > R is a gas constant (8.314 J mol$^{-1}$ K$^{-1}$) and T the absolute temperature.

L245: k1 number and units. Also, a reference should be cited here for adduct formation.

L276: with H-PPR and without H-PPR. > with H-PPR and without (w/o) H-PPR.

Evaluation of the impact of H-PPR on SOA Formation: aerosol acidity

L306: connected tothe > connected to the

Sensitivity of SOA formation to NOx 320 level, Temperature, and RH

L321: temperatures (278K, 288K and 298K) > temperatures (278 K, 288 K and 298 K)

L324: 2022 (between 6:30 AM to5:30… > 2022 (between 6:30 AM to 5:30…

L336: 2.82E-11 cm$^3$/molecule s$^{-1}$ at 298K >  …… and  [REFERENCE IS NEEDED}

L337: 1.2E-12 cm$^3$/molecule s$^{-1}$ >

Conclusion and atmospheric implications

L370: the heterogeneously produced PPR production occurs via…. > the H-PPR occurs via….

L371: OH radical > OH radicals that are

References

L444: New York2002

L460: doi [REMOVE UNDERLINE]

L464: New York2002

L489: doi [REMOVE UNDERLINE]

L493: with NO 2> with NO$_2$

L510: doi [REMOVE UNDERLINE]

L518: doi [REMOVE UNDERLINE]

L525: the absence of NOx, the absence of NO$_2$

L530: doi [REMOVE UNDERLINE]

L541: doi [REMOVE UNDERLINE]

L548: doi [REMOVE UNDERLINE]

L554: p-amino… > *p*-amino…

L563: doi [REMOVE UNDERLINE]/check doi

L572: doi [REMOVE UNDERLINE]

L580: *m*-xylene/doi [REMOVE UNDERLINE]

What were the wall deposition and the dilution rates for SOA, phenol, ozone and $NO_2$.

L624 and L625: 4:00 > 16:00 (as in Figs. 3 and 4)

L633: HO2 ….. RO2 > $HO_2$ ……$RO_2$

L639: Figure 2: the $k_2$ decomposition coefficient should be changed to match $k_{phenoxy}$ used in the manuscript body.

L642: k1 subscript

L645: Figure 3 and Figure 4 could be split into two parts, one for phenol and one for benzene. Try to use the same style (denotions, legend) for all the figures presented in Fig 3. and Fig. 4. Some left/right and down ticks would fit decently for all the figures. In Fig 3(B) and 3(C), Fig 3(H) and 3(I) use the same scale for better comparison. There is a different trend of HC in Fig 3 (A), (B), (C) in the first two hours compared with others for phenol. In Fig 3 (G-L) you have some variations. Are those in the uncertainty domain for your measurements? Fit the expiatory text of the experiment in the corresponding figure.

L646: Use subscripts for the inorganic species in figures and also in figure capitations!

L663: 2ppb > 2 ppb; L664: 298K > 298 K

L666 and L675: Same observations as in other figures. Figure 7 (C) scale, axis titles and legend color are different. Subscripts for OMAR and OMP to be consistent with the text. 298K > 298 K.

Supplement material

Stoichiometric coefficients

Please verify Eq. 13 > A1, B1, C1, D1 parameters!

Section 4: Check the subscripts for chemical compounds (i.e., H2O2) and superscript for units and large numbers. Italic font for notations (i.e., $k_{ph}$)

---

## Author Comment (AC1)

**Reviewer #1**

**General comment**. The manuscript "Dual roles of inorganic aqueous phase on SOA growth from benzene and phenol" provides new insight into the SOA formation processes from the oxidation of gaseous benzene and phenol under various HC:NO$_x$ ratios. To date, experimental studies show a negative related NO$_x$ dependence of SOA formation yield from the oxidation of aromatic hydrocarbons. The work presented herein combines experimental chamber investigations with a complex modeling system to deeply explore the heterogenous chemistry within SOA particles with respect to various relevant environmental parameters (i.e., acidity of SOA particles, SOA thermodynamical equilibrium, partitioning coefficients, temperature and RH). Authors employed a variety of modeling tools and used available atmospheric databases (MCM, EPI Suite) to design a tool for predicting the SOA mass under different atmospheric conditions by mean of heterogenous reactions in a two media particle system (inorganic/organic liquid phases) and by gas-particle partitioning processes. Acid-catalyzed formation of a persistent phenoxy radical (PPR) in wet inorganic aerosols and its desorption into the gas phase is hypothesized to be responsible for ozone consumption, thus lowering the atmospheric oxidation capacity near human settlements. Significant improvements were made to the in-use UNIPAR model by integrating HOM and H-PPR sequences to accommodate a new gas mechanism driven by the oxidation of benzene and phenol. Both the experimental and the modeling part are well presented through the manuscript. I recommend this manuscript for publication in ACP after the following concerns are addressed.

**Response to the general comment:** Thank you for your encouraging feedback on our paper. We have addressed your comments, to show clarity and better comprehension.

**Comment 1.** You may be more explicit in the abstract about the "Dual roles of inorganic aqueous phase". For instance, "Data presented herein highlights the impact of aqueous phase on SOA generated through benzene and phenol oxidation. The roles of the aqueous phase consist in: (1) and (2).

> **Response:** Clarify of dual role of aqueous phase on SOA formation was added in abstract, and reads now,

> "Benzene, emitted from automobile exhaust and biomass burning, is ubiquitous in ambient air. Benzene is a precursor hydrocarbon (HC) that forms secondary organic aerosols (SOA), but its SOA formation mechanism is not well studied.  To accurately predict the formation of benzene SOA, it is important to understand the gas mechanisms of phenol, which is one of the major products formed from the atmospheric oxidation of benzene. Our chamber study found that wet-inorganic aerosol (1) retarded the gas oxidation of phenol or benzene, and (2) suppressed SOA formation. To explain the unusual effect of aqueous phase, it is hypothesized that a PPR effectively forms via a heterogeneous reaction of phenol and phenol-related products in the presence of wet-inorganic aerosol. These PPR species are capable of catalytically consuming ozone during a $NO_x$ cycle and negatively influencing SOA growth.  In this study, explicit gas mechanisms were derived to produce the oxygenated products from the atmospheric oxidation of phenol and benzene. Gas mechanisms include the existing Master Chemical Mechanism (MCM v3.3.1); the reaction path for peroxy radical adducts originating from the addition of an OH radical to phenols forming low-volatility products (e.g., multi-hydroxy aromatics); and the mechanisms to form heterogeneous production of PPR. The simulated gas products were classified into volatility-reactivity based lumping species and incorporated into the UNIfied Partitioning Aerosol Reaction (UNIPAR) model that predicts SOA formation via multiphase reactions of phenol or benzene. The predictability of the UNIPAR model was examined using chamber data, which were generated for the photooxidation of phenol or benzene under various experimental conditions ($NO_x$ levels, humidity, and inorganic seed types). The SOA formation from both phenol and benzene still increased in the presence of wet inorganic seed because of the oligomerization of reactive organic species in aqueous phase. However, model simulations show a significant suppression in ozone, the oxidation of phenol or benzene, and SOA growth, compared to those without PPR mechanisms. In addition, the production of PPR is accelerated in the presence of acidic aerosol and this weakens SOA growth. In benzene oxidation, up to 53% of the oxidation pathway is connected to phenol formation in the reported gas mechanism. Thus, the contribution of PPR to gas mechanisms is less than phenol. Overall, SOA growth in phenol or benzene is negatively related to $NO_x$ levels in the high $NO_x$ region (HC ppbC/$NO_x$ ppb <5). However, the simulation indicates that the significance of PPR rises with decreasing $NO_x$ levels. Hence, the influence of $NO_x$ levels on the SOA formation from phenol or benzene is complex under varying temperature and seed types. The integration of comprehensive explicit gas mechanisms of phenolic compounds with SOA model will improve the prediction of SOA formation form the oxidation anthropogenic HCs and wildfires smoke."

**Comment 2.** A discussion regarding minimal incremental reactivity index (MIR) (Carter, 1994/ https://doi.org/10.1080/1073161X.1994.10467290) and photochemical ozone creation potentials (POCP) (Jenkin et al., 2017/ https://doi.org/10.1016/j.atmosenv.2017.05.024) of monocycle aromatics would add considerable impact to your current findings and highlight the atmospheric implications.

> **Response**: The relatively low MIR or POCP values of phenol and benzene can be explained via catalytic consumption of ozone by PPR, we cited paper recommended by the reviewer. This reads below equation R2 in Section 3.2.2 PPR formation in the revised manuscript.

**Comment 3.** To what extent would the competing reaction of PPR with the dissolved $NO_2$ in the inorganic phase affect the UNIPAR/H-PPR model (Kleffmann et al., 1998/ https://doi.org/10.1016/S1352-2310(98)00065-X)? Same question for the catechol gas-phase reactions with ozone (Obeid et al., 2024/ https://doi.org/10.1016/j.envpol.2023.122743; Coeur-Tourneur et al., 2009/ https://doi.org/10.1016/j.atmosenv.2008.12.054; Thomas et al., 2003/ https://doi.org/10.1002/kin.10121)

> **Response**: In our mechanism, the reaction of phenoxy radicals (phenol, catechol, and nitrophenols) with $NO_2$ (R3, R4, and R5) occurs mainly in the gas phase. In the ambient temperature, the partitioning of $NO_2$ into the inorganic salted aqueous phase is generally low. This reaction would possibly active in the stratospheric atmosphere, where temperature is low (average -50 $^{\circ}$C).

**Comment 4.** How is $k_{off\_phenoxy}$ calculated? Is it assumed to be equal to $k_{off\_phenol}$? If so, explain why and how an order of magnitude in between the considered value impact the model? Does the model incorporate Leighton equilibrium in predicting the gas-phase $O_3$, $NO_2$ and $NO$ concentrations?

> **Response**: Rate constant $k_{off\_phenoxy}$ was calculated differently from $k_{off\_phenol}$ due to their different chemical structure. For example, phenol can be both a donor and an accepter for hydrogen bonding but a phenoxy radical can be an accepter only. $k_{off}$ is calculated by Eq.5 which is influenced by $K_{in}$ and $k_{on}$. As seen in Eq.3, we assumed that the oxygen radical in the phenoxy radical is treated as a ketone functional group to calculate its vapor pressure and activity coefficient.

**Comment 5.** Kwok and Atkinson SAR on monocyclic aromatics follows the regression log $(k/cm^3\ molecule^{-1}\ s^{-1}) = -11.6 - 1.39\ \Sigma\sigma+$, where $\sigma+$ are the Hammett constants for electrophilic substitution by Brown and Okamoto (1958/ https://doi.org/10.1021/ja01551a055). If you are using EPI Suite software to estimate the gas /kinetic rate coefficients for multi-hydroxy benzenes

with vicinal OH groups the software may underestimate the values (Roman et al., 2022/ https://doi.org/10.5194/acp-22-2203-2022). '

> **Response**: We use the structure reactivity relationship using the group table. This estimation method can be reliable when it is used in its database, but extrapolation to organic compounds outside of the database results in a lack of the assurance of its accuracy. This reads in at the end of Section 3.2.1.
>
> "Brown, H. C., Okamoto, Y.: Electrophilic Substituent Constants, J. Am. Chem. Soc., 80, 4979-4987, 10.1021/ja01551a055, 1958"

**Comment 6.** Also, you could calculate and provide in the discussions sections a relative drop in $NO_2$, $O_3$ and SOA mass concentration when applying the UNIPAR with and without H-PPR.

> **Response**: The simulation of gas concentration was described in Fig.3 and discussed in Ln 275 to 279. SOA formation with or without H-PPR was discussed in Fig.4 and lines between 289 to 292.

**Comment 7.** Using the current dataset for the UNIPAR/H-PPR, could you estimate the SOA mass distribution from the oxidation of 2-methylphenol and catechol under similar conditions?

> **Response:** Yes, catechol and *o*-cresol are products of the oxidation of benzene (or phenol) and toluene, respectively. In this study, toluene was not included. The oxidation of *o*-cresol for the gas mechanism to form PPR was studied in the previous paper (Choi and Jang, 2022). The oxidation paths of benzene and toluene include the mechanisms of catechol and *o*-cresol.
>
> "Choi, J. and Jang, M.: Suppression of the phenolic SOA formation in the presence of electrolytic inorganic seed, Science of The Total Environment, 851, 158082, doi:10.1016/j.scitotenv.2022.158082, 2022."

**For technical corrections, minor questions and suggestions**

**Comment:** Affiliation is not indicated for the authors.

> **Response:** This has been done.

**Abstract**

**Comment** L10: gas oxidation or phenol or benzene… > gas oxidation of phenol or benzene…

**Response:** This has been done.

**Comment** L25: oxidation, about 53% of the… > oxidation, up to 53% of the…

**Response:** This has been done.

**Comment** Across the manuscript you have no consistency expressing the units (i.e., L227: g mol-1, L241: g/L). Choose one way to express the units.

**Response:** This has been done. The units system was unified.

**Introduction**

**Comment** L41: oxidation rate (i.e., 1.21571E-12 at 298K) > oxidation rate (i.e., $1.22 \times 10\text{-}12$ cm3 molecule-1 s-1 at 298K) [REFERENCE NEEDED]. Be consistent with the units and the order of magnitude across the manuscript and the supplement material. but its SOA yield is high > [provide a range for observed SOA formation yield and the corresponding cited paper/ papers].

**Response:** Reference for benzene oxidation rate was added. Sentence in L42 provides a citation regarding the benzene SOA yield.

"Borrás,E., Tortajada-Genaro, L. A.: Secondary organic aerosol formation from the photo-oxidation of benzene, Atmos. Environ., 47, 154-163, https://doi.org/10.1016/j.atmosenv.2011.11.020, 2012."

**Comment** L59: The lifetime is long also due to a p-π conjugated system also help for stabilizing the phenoxy radicals.

**Response:** The information was added.

**Comment** L85: delete "4-9, 52".

**Response:** This has been done.

**Comment** L86 of phenol or benzene > of phenol and benzene

**Response:** This has been done.

**Experiment Section**

**Comment** L 109: Specify the instrument and the operating conditions used to monitor the HCs concentration presented in Fig 3. What were the sensitivities and the corresponding relative uncertainties for NO/NO$_x$ (Villena et al., 2012/ https://doi.org/10.5194/amt-5-149-2012) and O$_3$ (Spicer et al., 2012/https://doi.org/10.3155/1047-3289.60.11.1353) photometers? In what extent these uncertainties would affect the experimental findings?

**Response:** It was added to a revised manuscript.

**Comment L 116:** Regarding the SOA seeds, were particle diameters the same for all experiments? Do you account for differences in SOA surface concentration in the UNIPAR model?

> **Response:** The geometric mean diameter of seeds is on average 146 nm. In the current UNIPAR model, the surface concentration of seed is not counted. SOA formation is calculated based on aerosol volume.

**Comment L 117**: sulfate, ammonium, nitrate ion peaks in aerosol. > sulfate, ammonium and nitrate ion signals in aerosol phase.

> **Response:** This has been changed.

**Comment L 120:** species (Sulfate, nitrate… > species (sulfate, nitrate…

> **Response:** This has been changed.

**UNIPAR SOA model**

**Comment L153:** You stated that "Both organic-phase oligomerization and aqueous reactions of reactive species in inorganic phase yield non-volatile OM in the model". Except for PPR, right?

> **Response:** This sentence is not related to PPR formation. The production of PPR influences the gas oxidation and the rate of production of oxygenated products. Ultimately, the retardation of gas oxidation slows down the SOA formation. This reads now in Section 3.1 2),
>
> "…The distribution of products was influenced by H-PPR as a function of the amount of sulfuric acid. H-PPR increases the contribution of fresh product distribution."

**HOM Formation**

**Comment L178**: The reaction rate constants > The gas phase reaction rate coefficients

> **Response:** We decided to use the term constant instead of coefficient since it includes only numbers.

**PPR Formation**

**Comment L183:** A citation needed for branching ratios.

**Response**: Citations for branching ratio have been added to reference of the revised manuscript.

"Jenkin, M. E., Saunders, S. M., Wagner, V., and Pilling, M. J.: Protocol for the development of the Master Chemical Mechanism, MCM v3 (Part B): tropospheric degradation of aromatic volatile organic compounds, Atmos. Chem. Phys., 3, 181–193, https://doi.org/10.5194/acp-3-181-2003, 2003."

"Bloss, C., Wagner, V., Jenkin, M. E., Volkamer, R., Bloss, W. J., Lee, J. D., Heard, D. E., Wirtz, K., Martin-Reviejo, M., Rea, G., Wenger, J. C., and Pilling, M. J.: Development of a detailed chemical mechanism (MCMv3.1) for the atmospheric oxidation of aromatic hydrocarbons, Atmos. Chem. Phys., 5, 641–664, https://doi.org/10.5194/acp-5-641-2005, 2005."

**Comment L222:** R is a gas constant (8.314 J mol-1 K-1). > R is a gas constant (8.314 J mol-1 K-1) and T the absolute temperature.

**Response:** This has been changed.

**Comment L245:** k1 number and units. Also, a reference should be cited here for adduct formation.

**Response:** We determined the rate constant of *phenol* (*in*) in reacts with the OH radical ($OH(in)$) in in phase to form an intermediate adduct of phenol (*phenol_OH_int* (*in*)) empirically. The sentence reads now,

"*Phenol* (*in*) in R6 further reacts with the OH radical ($OH(in)$) in *in* phase to form an intermediate adduct of phenol ($phenol\_OH\_int$ (*in*)) at reaction rate constant, $k_1$ (mol/L) which was determined empirically and very fast. Owing to the fast reaction, the reaction is limited by the concentration of OH radical."

**Comment L276:** with H-PPR and without H-PPR. > with H-PPR and without (w/o) H-PPR.

**Response:** This has been changed.

**Evaluation of the impact of H-PPR on SOA Formation: aerosol acidity**

**Comment L306:** connected tothe > connected to the

**Response:** Typo has been done.

**Sensitivity of SOA formation to NO_x 320 level, Temperature, and RH**

**Comment L321:** temperatures (278K, 288K and 298K) > temperatures (278 K, 288 K and 298 K)

    **Response:** The has been corrected.

**Comment L324:** 2022 (between 6:30 AM to5:30… > 2022 (between 6:30 AM to 5:30…

    **Response:** This has been corrected.

**Comment L336: REFERENCE** IS NEEDED

    **Response**: Citations have been added in reference and reads now,

    "Yee, L. D., Kautzman, K. E., Loza, C. L., Schilling, K. A., Coggon, M. M., Chhabra, P. S., Chan, M. N., Chan, A. W. H., Hersey, S. P., Crounse, J. D., Wennberg, P. O., Flagan, R. C., and Seinfeld, J. H.: Secondary organic aerosol formation from biomass burning intermediates: phenol and methoxyphenols, Atmos. Chem. Phys., 13, 8019–8043, https://doi.org/10.5194/acp-13-8019-2013, 2013."

    "Kwok, E. S.C., Atkinson, R.: Estimation of hydroxyl radical reaction rate constants for gas-phase organic compounds using a structure-reactivity relationship: An update, Atmos. Environ. 29, Issue 14, 1685-1695, https://doi.org/10.1016/1352-2310(95)00069-B, 1995."

**Comment L337:** 1.2E-12 cm3/molecule s-1 >

    **Response:** The unit has been changed in unified system.

**Conclusion and atmospheric implications**

**Comment L370:** the heterogeneously produced PPR production occurs via…. > the H-PPR occurs via….

    **Response:** This has been changed.

**Comment L371:** OH radical > OH radicals that are

**Response:** This has been changed.

**Comment:** What were the wall deposition and the dilution rates for SOA, phenol, ozone and NO$_2$.

> **Response:** Wall loss factor and dilution rate for SOA, hydrocarbon vapor, and gases were obtained from experimental data. Information were added to footnote of table 1 and reads now,

> e. "The reported SOA mass was corrected for the particle loss to the chamber wall based on the 1$^{st}$ order deposition rate at 64 particle size beans. The dilution rate of SOA is estimated with the gas dilution factor determined using trace gas (CCl$_4$)."

**Comment** L624 and L625: 4:00 > 16:00 (as in Figs. 3 and 4)

> **Response:** This has been changed.

**Comment L633**: HO2 ….. RO2 > HO$_2$ ……RO$_2$

> **Response:** This has been changed.

**Comment L639**: Figure 2: the k2 decomposition coefficient should be changed to match kphenoxy used in the manuscript body.

> **Response:** This has been changed.

**Comment L642**: k1 subscript

> **Response:** This has been changed.

**Comment L645**: Figure 3 and Figure 4 could be split into two parts, one for phenol and one for benzene. Try to use the same style (denotions, legend) for all the figures presented in Fig 3. and Fig. 4. Some left/right and down ticks would fit decently for all the figures. In Fig 3(B) and 3(C), Fig 3(H) and 3(I) use the same scale for better comparison. There is a different trend of HC in Fig 3 (A), (B), (C) in the first two hours compared with others for phenol. In Fig 3 (G-L) you have some variations. Are those in the uncertainty domain for your measurements? Fit the expiatory text of the experiment in the corresponding figure.

> **Response:** We decided to keep original figures without splitting. For Fig 3(B), (C), (H), and (I), the same scale was used in the revised manuscript.

**Comment L646:** Use subscripts for the inorganic species in figures and also in figure capitations!

     **Response:** This has been done.

**Comment L663:** 2ppb > 2 ppb; L664: 298K > 298 K

     **Response:** This has been done.

**Comment L666 and L675**: Same observations as in other figures. Figure 7 (C) scale, axis titles and legend color are different. Subscripts for OMAR and OMP to be consistent with the text.

     **Response:** This has been done.

**Supplement material**

**Stoichiometric coefficients**

**Comment :** Please verify Eq. 13 > A1, B1, C1, D1 parameters!

     **Response:** This has been done.

**Comment on Section 4:** Check the subscripts for chemical compounds (i.e., $H_2O_2$) and superscript for units and large numbers. Italic font for notations (i.e., kph)

     **Response:** This has been done.

**Comment on References**

L444: New York2002

L460: doi [REMOVE UNDERLINE]

L464: New York2002

L489: doi [REMOVE UNDERLINE]

L493: with NO 2> with $NO_2$

L510: doi [REMOVE UNDERLINE]

L518: doi [REMOVE UNDERLINE]

L525: the absence of $NO_x$, the absence of $NO_2$

L530: doi [REMOVE UNDERLINE]

L541: doi [REMOVE UNDERLINE]

L548: doi [REMOVE UNDERLINE]

L554: p-amino… > *p*-amino…

L563: doi [REMOVE UNDERLINE]/check doi

L572: doi [REMOVE UNDERLINE]

L580: *m*-xylene/doi [REMOVE UNDERLINE]

    **Response:** These have been done.

---

## Author Comment (AC2)

Manuscript: egusphere-2023-2461

**Reviewer #2**

**General comment**. The study by Jiwon Choi et al. "Dual roles of inorganic aqueous phase on SOA growth from benzene and phenol" is a combination of experimental and modeling data that give an insight on oxidation of gaseous benzene and phenol and the formation of secondary organic aerosol. In this work the authors showed negative relation of SOA growth (of phenol and benzene) to $NO_x$ levels in high $NO_x$ regions by using several databases and models. Furthermore, the simulations in the current work showed the aspect of increasing significance of persistent phenoxy radical with decreasing $NO_x$ levels. Significances of persistent phenoxy radicals that are produced during wildfire plumes and their impact on retarding the atmospheric oxidation in urban areas are highlighted. The manuscript is well written and fits well in the scientific scope of ACP. I recommend the manuscript to be published after some minor revision.

**Response to the general comment:** We would like to thank the reviewer for the time and the constructive comments on our manuscript. The comments are reproduced below along with the author response. Any change made in the manuscript is in the track change mode and that in the supporting information in the blue color.

**Minor revision:**

**Comment Line 63:** I suggest to add an author to the UNIPAR model: https://doi.org/10.5194/acp-14-4013-2014 .

> **Response**: Author of the UNIPAR model was added to the revised manuscript.

**Comment Line 77**: You might specifiy: "In this study, we hypothesize [based on chamber experiments and complex model data] that the production …".

> **Response**: This has been done.

**Comment Line 85**: citation style changed or the numbers need to be deleted.

> **Response**: The numbers have been deleted.

**Comment Line 152**: Do you also mean to include the persistent phenoxy radicals or not? "Both organic-phase oligomerization and aqueous reactions of reactive species in inorganic phase yield non-volatile OM in the model".

> **Response**: Reactive species partitioned into both organic phase and inorganic phase can be oligomerized to form non-volatile organic species increasing SOA. In particular, oligomerization is catalyzed by acid in inorganic aqueous phase. The description of the UNIPAR model does not include PPR formation. The production of PPR is described in

Section 3.2.2 PPR Formation. The formation of PPR influences gas oxidation and the rate of production of oxygenated products. Ultimately, the retardation of gas oxidation can slow down the SOA formation predicted by the UNIPAR model. This reads now in Section 3.1 2),

"…The distribution of products was influenced by H-PPR as a function of the amount of sulfuric acid. H-PPR increases the contribution of fresh product distribution."

**Comment Line 183**: There is a citation/reference missing for the values in the brackets.

**Response**: Citations for branching ratio have been added to reference of the revised manuscript.

"Jenkin, M. E., Saunders, S. M., Wagner, V., and Pilling, M. J.: Protocol for the development of the Master Chemical Mechanism, MCM v3 (Part B): tropospheric degradation of aromatic volatile organic compounds, Atmos. Chem. Phys., 3, 181–193, https://doi.org/10.5194/acp-3-181-2003, 2003."

"Bloss, C., Wagner, V., Jenkin, M. E., Volkamer, R., Bloss, W. J., Lee, J. D., Heard, D. E., Wirtz, K., Martin-Reviejo, M., Rea, G., Wenger, J. C., and Pilling, M. J.: Development of a detailed chemical mechanism (MCMv3.1) for the atmospheric oxidation of aromatic hydrocarbons, Atmos. Chem. Phys., 5, 641–664, https://doi.org/10.5194/acp-5-641-2005, 2005."

**Comment Line 336/337**: There is a citation/reference missing for the values in the brackets.

**Response**: Citations have been added in reference and reads now,

"Yee, L. D., Kautzman, K. E., Loza, C. L., Schilling, K. A., Coggon, M. M., Chhabra, P. S., Chan, M. N., Chan, A. W. H., Hersey, S. P., Crounse, J. D., Wennberg, P. O., Flagan, R. C., and Seinfeld, J. H.: Secondary organic aerosol formation from biomass burning intermediates: phenol and methoxyphenols, Atmos. Chem. Phys., 13, 8019–8043, https://doi.org/10.5194/acp-13-8019-2013, 2013."

"Kwok, E. S.C., Atkinson, R.: Estimation of hydroxyl radical reaction rate constants for gas-phase organic compounds using a structure-reactivity relationship: An update, Atmos. Environ. 29, Issue 14, 1685-1695, https://doi.org/10.1016/1352-2310(95)00069-B, 1995."

**Comment Figure S1**: Please check the caption of the second y-axis.

**Response**: Typo has been corrected.

**General comments:**

**Comment:** Missing affiliation indication for authors.

   **Response**: Author affiliation indications were added.

**Comment:** check the subscripts for chemical compounds in the text, figures and figure capitations; check the consistency of the units in the manuscript and remain with one style.

   **Response**: Subscripts were checked. Unit styles were unified.

**Comment on Reference Section:** please remove the lines under the DOI link.

   **Response**: This has been done.

---

## Author Comment (AC3)

**Reviewer #3**

**General comment**. The manuscript "Dual roles of inorganic aqueous phase on SOA growth from benzene and phenol" provides coupled experimental and modelling evidence of the suppression of atmospheric oxidation capacity and SOA growth due to the formation of persistant peroxy radicals formed during the production of benzene and phenol derived SOA. The Heterogeneous Persistant Phenoxy Radical Model was derived with a new explicit mechanism for the formation of HOM and H-PPR and utilised in the UNIPAR model to predict the formation of SOA from multiphase reactions of phenol and benzene. The addition of H-PPR into model was found to increase suppression of SOA growth with this suppression found to further increase with increasing aerosol acidity.

**Response to the general comment:** Thank you for your thoughtful comments on this manuscript. Due to your comment, the quality of this paper has been improved much.

**Comment 1.** While the importance of this work on urban areas specifically is mentioned, I don't feel this is quantitively explored enough in the implications section. I would suggest restructuring this section as the conclusion currently reads more like a discussion to me with new ideas still being introduced (i.e. ln 406" Phenol is the most abundant first-generation product from the oxidation of benzene…") and a lot of use of generalizations in language such as "about" or "generally". Slightly more quantitative implications would help to cement the importance of this work. What urban areas of the world is this most likely to effect? Areas with more biomass burning and wildfires or areas such as Chinese megacities and haze dominated regions?

> **Response:** The section for Conclusion and Atmospheric Implication has been reconstructed to provide better flow and quantitative interpretation, and the revised modified paragraphs reads now,
>
> "Fundamentally, biomass burning under open flame is performed at low temperature and produces very low NO (Simoneit, 2002; Mebust and Cohen, 2013; Xu et al., 2021). The chemistry slows to a standstill without $NO_x$ and thus halts ozone formation although gaseous HCs are abundant. When these fire plumes mix into urban atmospheres abundant in $NO_x$, ozone formation becomes active, impacting the air quality of the city. Chamber data of this study mimics the phenol oxidation in the presence of $NO_x$. In addition, hygroscopic inorganic aerosols comprising of nitrate, sulfate and ammonium ions are available in the city environment rich in $NO_x$, $SO_2$ and $NH_3$. When wildfire plumes mix in city air, their phenolic compounds interact with $NO_x$ and hygroscopic inorganic aerosol. The results from this study suggest that PPR produced during the atmospheric process of phenolic compounds in wildfire plumes can retard the atmospheric oxidation in urban environments. The SOA simulation with the low concentrations of phenol and typical atmospheric tracer gas (formaldehyde, acetaldehyde) in Fig. 5 shows that phenol SOA is considerably suppressed even with a small amount of wet inorganic aerosol

raging from weakly to neutral acidity. For example, phenol SOA mass decreases by 12% with 5 ppb of ammonium hydrogen sulfate (FS=0.5) (Fig. 5(A)) and the SOA mass from the mixture of phenol and benzene decreased by 28% (Fig. 5(C)).

The impact of $NO_x$ on SOA formation appeared to be negative as shown in Fig. 6 under high $NO_x$ levels. A significant fraction of phenolic SOA is through HOM products and oligomeric matter. The contribution of HOM and oligomeric matter on SOA formation is generally higher with lower $NO_x$ levels. Thus, phenol SOA and benzene SOA are relatively insensitive to temperature (Fig. 6) due to the high fraction of SOA mass being non-volatile. This result suggests that SOA from biomass burning is not substantially affected by temperature under low $NO_x$ regimes. When the concentrations of $NO_x$ drop in the high $NO_x$ zone, SOA formation increases. The role of PPR on atmospheric oxidation capacity in the blending of wildfire smoke and urban pollutants needs to be studied under different $NO_x$ levels.

A variety of phenolic compounds including phenol, cresol, catechol, methoxyphenols, dimethylphenols (Akherati et al., 2020; Bruns et al., 2016) can consist of more than 80% of the precursor HCs in wildfire smoke. These multifunctional phenolic compounds can also yield PPR as active scavengers for ozone (Section 3.2.2). To date, the impact of phenolic compounds on retardation of atmospheric aging of HCs in the city air has not been sufficiently studied. It is important to comprehend the formation mechanisms of PPR-like chemical species and their role on atmospheric oxidation capability to accurately predict the elevation of ozone and SOA and their peak time."

**Comment 2.** Section 4.1 – suggest slight restructuring/rewording for increased clarity as it is a bit difficult to follow at present.

> **Response:** The Section 4.1 was reconstructed based on the comment, see in manuscripts Ln. 270 to Ln.286, reads now:
>
> "Fig. 3 (A-L) shows gas simulations (phenol, benzene, ozone, NO, and $NO_2$) based on data collected in the UF-APHOR chamber. In the presence of wet-inorganic seed (i.e., wet-AS, AHS, and SA), both simulations and chamber data demonstrate a notable suppression in gas oxidation, specifically in ozone formation and the decay of phenol or benzene, compared to gas oxidation in non-seeded conditions. Fig. 3(B) and (H) show gas simulations for phenol and benzene oxidation in the presence of SA seed without the H-PPR mechanism, revealing a significant discrepancy between simulations and observations. Fig. 3(C) and (I) demonstrate enhanced gas simulations using H-PPR under identical experimental settings, highlighting the significance of H-PPR in precisely forecasting the oxidation of phenol or benzene in the presence of wet inorganic aerosol, as outlined in reactions R3-R5. Aside from phenol, catechols and nitrophenols, which are significant byproducts of phenol oxidation, can also participate in the PPR formation. Suppressed ozone levels can decrease the generation of OH radicals and slow down the aging of organic substances. Explicit gas simulations incorporating HOM and H-PPR show good agreement with observations.

Fig. 4(A-P) shows chamber-generated SOA mass from the photooxidation of phenol (Fig. 4(A-H)) or benzene (Fig.4(I-P)) under various inorganic seed conditions (Table 1), along with simulations of SOA formation using the UNIPAR model. Overall, an enhanced SOA simulation of phenol or benzene was conducted using precise gas simulation combined with HOM and H-PPR. Fig. 4(A) and (B) display non-seed phenol SOA, while Fig. 4(I) and (J) show non-seed benzene. Fig. 4 (C-H) displays SOA masses generated with inorganic seed (SA, AHS, wet- or dry-AS) for phenol, while Fig. 4 (K-P) shows the masses for benzene.

Fig. 4(C) and (G) show the significance of H-PPR mechanisms in predicting SOA for phenol, while Fig. 4(K) and (O) demonstrate this for benzene by comparing simulations with and without H-PPR. The suppression of SOA formation was greater with highly acidic aerosol.

The formation rate of PPR can be affected by the chemical composition of the aerosol medium. Mitroka et al. (Mitroka et al., 2010) reported that reactivity of the OH radical is considerably higher in polar, protic solvent than that in dipolar, aprotic solvent. Protic solvent is a hydrogen bond donor that stabilizes the transition state of the OH radical addition reaction. Thus, the reaction of phenols with the OH radical is more favorable in in phase than or phase. The radical scavenging ability of phenols by forming phenoxy radicals is in the order of pyrogallol > 1,2,4-benzenetriol >catechol > hydroquinone > resorcinol ≈ phloroglucinol (Thavasi et al., 2009). As shown in reaction R8, phenol in salted aqueous media reacts with OH(in) in a similar way with the OH addition to the aromatic ring in the gas phase to form intermediate product phenol_OH_int (in) (Fig. 2). Fig. S3 is the proposed mechanism to form phenoxy radical via the acid-catalyzed reaction. In addition, some organic products such as quinones can promote increased oxidants in aqueous acidic media. Quinones are well recognized for their ability to promote superoxide formation (Guin et al., 2011). Lowering pH increases the redox potential (Walczak et al., 1997) of quinone-hydroquinone. However, the reduction potential of oxygen can be lower in acidic condition and is advantageous for $O_2\bullet^-/HO_2\bullet$ formation (Wei et al., 2022) (Section S4).

The importance of HOM on phenol SOA has been demonstrated in the previous study by Choi and Jang (Choi and Jang, 2022). For example, a large fraction of OMP in Fig. 4(A) is contributed by HOM. The contribution of HOM to SOA mass increases with decreasing NO levels. The systematic evaluation of the UNIPAR model integrated with the explicit gas mechanisms will be performed via the model sensitivity to various environmental variables (i.e., $NO_x$ levels, seed, temperature, and humidity) in Section 4.2."

**Comment 3.** Adding in the factors of suppression for the different model scenarios may help to add context to level of suppression of SOA growth exhibited. At present, throughout the manuscript this is not directly given a number.

> **Response:** We added the suppression SOA formation with and without H-PPR in Fig. 5. Please find the response to comment 1.

**Comment 4.** Have you considered natural emissions of benzene and phenol such as over polar oceans or in the marine boundary layer? (Wohl et al., 2023 Sci. Adv.)

> **Response:** Thank you for pointing out the flux of benzene from oceans or in the marine boundary layer. We cited the recommended paper in introduction.

**Comment 5.** How competitive is $NO_3$ oxidation of phenol to give $C_6H_5O$ compared to the reactions with OH and $O_3$? Is this significantly fast as a dark reaction to be considered important? When $NO_3$ is the oxidiser is the oxidation capacity of the system still suppressed with increasing aerosol acidity?

> **Response:** Thanks for the comment on $NO_3$ radical. In order to response to the reviewer's comment, we perform the integrated reaction rate (IRR) analysis. The IRR analyses demonstrate that (1) H-PPR production is important to form PPR in the presence of wet seed (description in Section 4.2.1) and (2) the nitrate radical oxidation with phenol is important to form PPR during nighttime (description in Section 4.2.2). This reads now,
>
> Section 4.2.1 (2nd paragraph)
>
> "The DISMACC box model flatform of this study is equipped with the integrated reaction rate (IRR) analysis technique, which can show the chemical reaction flow in the oxidation mechanisms. Based on the IRR analysis, the production of PPR is mainly contributed by the reaction of phenol with OH radicals in gas phase and the catechol H-PPR mechanism. The phenoxy radical production via the phenol H-PPR path is trivial due to the low partitioning of phenol into the aqueous phase. Unlike phenol, catechol, a major product of phenol oxidation, can yield the semiquinone radical (PPR of catechol) via the heterogeneous reaction mechanism. For example, in the presence of SA seed (Fig. 5(A)), the contribution of the catechol H-PPR path is 1.22 times greater than that of the gas-phase reaction of phenol with OH radicals. In the presence of wet-AS seed, the contribution of the catechol H-PPR path is 20% of that from the gas-phase reaction of phenol with OH radicals."
>
> Section 4.2.2 (4th paragraph)
>
> "Of the total PPR production, the contribution of daytime phenol oxidation with nitrate radicals is only 0.1% of that from the phenol oxidation with OH radicals in the high $NO_x$ condition (VOC ppbC/$NO_x$ ppb = 2) of Fig. 6(B). However, the contribution of nitrate radical mechanism to form PPR increases in the absence of sunlight. For example, the contribution of nitrate radicals on PPR is nearly 30% of that with OH radicals at a given simulation condition under the high $NO_x$ condition between 4PM to 5PM (Fig. 6(A)). This simulation result suggests that the nitrate radical oxidation with phenol is important to form PPR during nighttime."

**Minor corrections:**

**Comment 1.** Throughout the manuscript the consistency of inclusion of units and unit formatting needs to be checked.

    **Response:** Units and formatting have been corrected thoroughly.

**Comment 2.** Throughout the manuscript there is repeated mentions of other phenolic compounds. In ln 281 it is mentioned that "The radical scavenging ability of phenols by forming phenoxy radicals is in the order of pyrogallol > 1,2,4- benzenetriol >catechol > hydroquinone > resorcinol ≈ phloroglucinol". This being said, why was the focus of the study not expanded to include some of the more active phenols, especially as phenol can form.

    **Response:** Phenol is a starting precursor of this study. Phenol oxidation yields the multi-hydroxy substitute phenols, and their product can produce PPR via H-PPR mechanisms. For example, the catechol H-PPR mechanism is included in the gas mechanism in addition to phenol. The gas mechanism also produces pyrogallol, but this concentration is trivial.

**Comment 3.** Consistency in nomenclature is needed. For example pyrogallol and catechol are not given in chemical nomenclature, but 1,2,4-benzenetriol is. Suggest changing to hydroxyhydroquinone or changing the others, ie pyrogallol to 1,2,3-benzenetriol, or catechol to 1,2- benzenediol.

    **Response:** The nomenclature of multi hydroxy benzenes has been corrected using the IUPAC guideline except catechol (1,2-dihydroxybenzene).

**Comment 4**. I suggest adopting a consistent colour for benzene and for phenol in all figures. Being different colours in every figure reduces readability.

    **Response:** Colors in Figs.3, 4, and 8 have been changed.

**Technical revisions:**

**Abstract:**

**Comment 1:** Missing author affiliation numbers. Add in what the implication of this research is atmospherically.

    **Response:** This has been added.

**Comment 2:** Is the mention of $NO_x$ limited regimes worth highlighting more?

    **Response:** The $NO_x$ regime originates from the relative importance of the reaction between $NO_2$ and OH compared to the reaction between hydrocarbons and OH radical. In general, the reaction of $NO_2$ with OH radical is more important than the reaction of hydrocarbon with OH in the region of the VOC ppbC/$NO_x$ ppb ratio less than 5. In the

high $NO_x$, OH radical is consumed by $NO_2$ and the products include more organonitrate and PANs, which can produce SOA by the gas-particle partitioning process. In addition, HOM production is high in the low $NO_x$ region.

**Comment 3:** Add in what the implication of this research is atmospherically.

**Response:** This has been added.

**Introduction:**

**Comment1 :** Add in some statistics and refs of the prevalence of benzene and phenol atmospherically.

**Response:** The emission of phenol from biogenic burning is included with the reference, and reads now,

"In addition, a significant fraction (20%) of oxygenated aromatic, emitted from biomass burning, is phenol (Akherati et al, 2020)."

**Comment 2:** Ln 35 – where globally is represented by 20 % and where 90 %?

**Response:** Citation has been added and reads now in the refence of the revised manuscript,

**"**Kanakidou, M., Seinfeld, J. H., Pandis, S. N., Barnes, I., Dentener, F. J., Facchini, M. C., Van Dingenen, R., Ervens, B., Nenes, A., Nielsen, C. J., Swietlicki, E., Putaud, J. P., Balkanski, Y., Fuzzi, S., Horth, J., Moortgat, G. K., Winterhalter, R., Myhre, C. E. L., Tsigaridis, K., Vignati, E., Stephanou, E. G., and Wilson, J.: Organic aerosol and global climate modelling: a review, Atmos. Chem. Phys., 5, 1053–1123, https://doi.org/10.5194/acp-5-1053-2005, 2005."

**Comment 3:** Ln 45 – references needed

**Response: Citation has been added into the revised manuscript.**

**"**Borrás,E., Tortajada-Genaro, L. A.: Secondary organic aerosol formation from the photo-oxidation of benzene, Atmos. Environ., 47, 154-163, https://doi.org/10.1016/j.atmosenv.2011.11.020, 2012."

**Comment 4:** Where is the benzene oxidation path important? Why is it hard to study benzene oxidation specifically?

**Response:** The benzene oxidation rate is relatively very slow ($1.22 \times 10\text{-}12$ cm3 molecules-1 sec-1 at 298 K). Therefore, the consumption of the benzene is little and SOA formation is low.

**Comment 5:** Expand the mention of wildfire SOA to mention briefly atmospheric implications.

    **Response:** In the 3$^{rd}$ and 5$^{th}$ paragraphs in Section "5. Conclusion and Atmospheric Implications", wildfire SOA has been discussed.

**Comment 6:** Ln 66- what relative humidity are you defining as "wet" and is it consistent over all experiments?

    **Response**: The definition of wet is determined based on the inorganic phase state. The wet inorganic aerosol contains water, but dry aerosol is efflorescent with no water. The UNIPAR model calculates efflorescence relative humidity (ERH) and water content and applies aerosol chemistry in the aqueous phase. Please find model description Section 3.1.

**Comment 7:** Ln 69 – add more recent references for aerosol acidity experiments

    **Response:** More references have been added in the revised manuscript, reads now,

    "Deng, Y., Inomata, S., Sato, K., Ramasamy, S., Morino, Y., Enami, S., and Tanimoto, H.: Temperature and acidity dependence of secondary organic aerosol formation from α-pinene ozonolysis with a compact chamber system, Atmos. Chem. Phys., 21, 5983–6003, https://doi.org/10.5194/acp-21-5983-2021, 2021."

    "Surratt, J. D., Lewandowski, M., Offenberg, J. H., Jaoui, M., Kleindienst, T. E., Edney, E. O., and Seinfeld J. H.: Effect of Acidity on Secondary Organic Aerosol Formation from Isoprene. Environ. Sci. Technol., 41, 5363-5369, DOI: 10.1021/es0704176, 2007."

**Model description:**

**Comment 1:** Ln 131 – typo, correct to UNIPAR

    **Response:** This has been corrected.

**Comment 2:** Figure 2 is too cramped with the ending 'e' of 'intermediate' cut off

    **Response:** This has been corrected.

**Results and Discussion (Section 4.1):**

**Comment 1:** Figure 3 – none of the axes are labelled or given units, these also need to be added to the figure caption.

**Response:** The unit of the axes is labeled in fig 3 (A) only. For better readability, the figure caption has been changed and reads now,

"The time profiles of observations and the prediction for concentrations of NO, $NO_2$, and $O_3$ and hydrocarbons (Table 1). The x-axis represents time (EST), the first y-axis represents the concentration (ppb) of gas species, and the second y-axis represents the hydrocarbon concentration in gas phase (ppb) as shown in (A). "HC" and "HC_exp" demote the gas simulation of hydrocarbons used in experiment and measurement of hydrocarbon used in experiment, respectively. The error associated with NO, $NO_2$, and $O_3$ are 2% and not visible in this Figure."

**Comment 2**: Both Figure 3 and 4 need to be reworked to help readability. I suggest splitting the figure into benzene and phenol, or perhaps phenol have dashed lines and benzene filled. At present the figure is hard to digest. The figure titles are also overpowering and the legend is too small and needs reordering so the predictions and corresponding experiments are side by side.

**Response**: We decided not to split the figures 3 and 4 into benzene and phenol. By this way, reader can compare benzene and phenol.

**Comment 3:** Ln 264 –How is improved quantified? What is this in relation to?

**Response:** The model simulation is evaluated by comparing predictions with chamber data. In order to clarify this, the sentence has been rewritten, and reads now in the 2nd paragraph of Section 4.1,

"Under the same experimental conditions, simulations with H-PPR in Fig. 3(C) and (I) well predict chamber data showing the importance of H-PPR (reactions R3-R5)."

**Comment 4**: Ln267 – should this be explained explicitly earlier as an indirect amplification of the scavenging of $O_3$?

**Response**: Section 4.1 has been reorganized to provide better flow. Please find the response to comment 2 (major comment).

**Comment 5**: Ln 272 – "accurate gas simulation" – why is it defined as accurate?

**Response**: Word "accurate" has been changed to "improved".

**Comment 6:** Ln 277 – how are you defining "highly acidic aerosol"?

**Response**: Word "highly" has been removed.

**Comment 7:** Ln 278 to 283 - feel a bit out of place.

**Response**: This has been done.

**Comment 8:** Ln 286 -289 – unclear why this is said here.

> **Response**: This explains the multi-functional phenols from phenol oxidation can promote the oxidants in aqueous phase, which could advantage the H-PPR mechanism.

**Comment 9:** Figure 7 – axis C label and legend font colour should be black not grey. And plot A is a side-by-side boxplot while B and C are not.

> **Response**: This has been done.

**Results and Discussion (Section 4.2):**

**Comment 1:** Ln 297 – Is "FS value" defined previously? If not, define here.

> **Response**: This has been defined before in Section 3.2.2.

**Comment2 :** Ln 299 – insert reference for deliquescence point of seed aerosol

> **Response:** The reference has been added in the revised manuscript, reads now,
>
> Peng, C., Chen, L., Tang, M.: A database for deliquescence and efflorescence relative humidities of compounds with atmospheric relevance, Fundamental Research, 2, 578-587, https://doi.org/10.1016/j.fmre.2021.11.021, 2022."

**Comment 3:** Ln 308 – why do you think that a larger conc of inorganic seed suppresses SOA mass? – for a given RH there is less water per seed particle? i.e. each surface has a thinner liquid microlayer?

> **Response:** The concentration of organic species in inorganic phase ($C_{in,i,}$) increase with the mass concentration of inorganic mass ($M_{in}$). $M_{in}$ increase with more inorganic aerosol mass and high humidity as seen in equation below.
>
> $$C_{in,i} = \frac{K_{in}M_{in}}{1+K_{or,i}OM_T+K_{in,i}M_{in}} C_{T,i} \text{ (Eq.S9)}$$
>
> This equation has been added into the Section "S2 UNIPAR Model Structure" of SI.

**Comment 4:** Ln 325 – why was 30 ppb chosen for the initial concentration?

> **Response:** This was set to mimic real world environment.

**Comment 5:** Ln 336 – 2.82E-11 reformat as 2.82 10-11 and provide reference

> **Response:** Citation has been added and reads now in reference,

"Yee, L. D., Kautzman, K. E., Loza, C. L., Schilling, K. A., Coggon, M. M., Chhabra, P. S., Chan, M. N., Chan, A. W. H., Hersey, S. P., Crounse, J. D., Wennberg, P. O., Flagan, R. C., and Seinfeld, J. H.: Secondary organic aerosol formation from biomass burning intermediates: phenol and methoxyphenols, Atmos. Chem. Phys., 13, 8019-8043, 10.5194/acp-13-8019-2013, 2013."

**Comment 6:** Ln 227 – 1.2E-12 reformat as 2 10-12 and provide reference

**Response:** Citation has been added and reads now in reference,

"Kwok, E. S.C., Atkinson, R.: Estimation of hydroxyl radical reaction rate constants for gas-phase organic compounds using a structure-reactivity relationship: An update, Atmos. Environ. 29, Issue 14, 1685-1695, https://doi.org/10.1016/1352-2310(95)00069-B, 1995."

**Comment 7:** Ln 336 to 340 – Is this repeated information, is it needed here?

**Comment 8** :Ln 350, Section 4.2.3 – is this section not also testing the senstivitiy, as opposed to the uncertainty?

**Response**: To clarify, section number has been changed. In the revised manuscript, sensitivity test in under Section 4.2 and uncertainty test is under Section 4.3.

**Comment 9**: Ln 355 – only the sensitivity of the benzene simulations are included here. What not also Phenol?

**Response**: Thank you for pointing out this. The missing description has been added to the revised manuscript, and reads now in Section "4.3 Uncertainty of SOA Formation to Model Parameters",

"Under the same condition, the change in the phenol SOA mass due to VP uncertainties ranges from -13% to 14%."